# Visualization of currents in neural models with similar behavior and different conductance densities

**Leandro M Alonso\*, Eve Marder**

Volen Center and Biology Department, Brandeis University, Waltham, United States

**Abstract** Conductance-based models of neural activity produce large amounts of data that can be hard to visualize and interpret. We introduce visualization methods to display the dynamics of the ionic currents and to display the models' response to perturbations. To visualize the currents' dynamics, we compute the percent contribution of each current and display them over time using stacked-area plots. The waveform of the membrane potential and the contribution of each current change as the models are perturbed. To represent these changes over a range of the perturbation control parameter, we compute and display the distributions of these waveforms. We illustrate these procedures in six examples of bursting model neurons with similar activity but that differ as much as threefold in their conductance densities. These visualization methods provide heuristic insight into why individual neurons or networks with similar behavior can respond widely differently to perturbations.

DOI: https://doi.org/10.7554/eLife.42722.001

## Introduction

Experimental and computational studies have clearly demonstrated that neurons and circuits with similar behaviors can, nonetheless, have very different values of the conductances that control intrinsic excitability and synaptic strength. Using a model of the crustacean stomatogastric ganglion (STG), *Prinz et al. (2004)* showed that similar network activity can arise from widely different sets of membrane and synaptic conductances. Recent experimental measurements have shown two- to six-fold variability in individual components in the same identified neurons (*Schulz et al., 2006*; *Schulz et al., 2007*; *Roffman et al., 2012*; *Swensen and Bean, 2005*). The use of RNA sequencing and other molecular measurements have shown significant cell-to-cell variability in the expression of ion channels (*Temporal et al., 2012*; *Temporal et al., 2014*; *Tobin et al., 2009*). Together these results suggest that similar activities arise from different cellular and network mechanisms. Here, we use conductance-based models to explore how different these mechanisms are and how they respond to perturbation.

Because of the intrinsic variability, canonical models that capture the mean behavior of a set of observations are not sufficient to address these issues (*Golowasch et al., 2002*; *Balachandar and Prescott, 2018*). To incorporate intrinsic biophysical variability *Prinz et al. (2004)* introduced an ensemble modeling approach. They constructed a database with millions of model parameter combinations, analyzed their solutions to assess network function, and screened for conductance values for which the activity resembled the data (*Calabrese, 2018*). An alternative was used by *Achard and De Schutter (2006)*. They combined evolutionary strategies with a fitness function based on a phase-plane analysis of the models' solutions to find parameters that reproduce complex features in electrophysiological recordings of neuronal activity, and applied their procedure to obtain 20 very different computational models of cerebellar Purkinje cells. Here, we adopt a similar approach and

\*For correspondence:
lalonso@brandeis.edu

**eLife digest** The nervous system contains networks of neurons that generate electrical signals to communicate with each other and the rest of the body. Such electrical signals are due to the flow of ions into or out of the neurons via proteins known as ion channels. Neurons have many different kinds of ion channels that only allow specific ions to pass. Therefore, for a neuron to produce an electrical signal, the activities of several different ion channels need to be coordinated so that they all open and close at certain times.

Researchers have previously used data collected from various experiments to develop detailed models of electrical signals in neurons. These models incorporate information about how and when the ion channels may open and close, and can produce numerical simulations of the different ionic currents. However, it can be difficult to display the currents and observe how they change when several different ion channels are involved.

Alonso and Marder used simple mathematical concepts to develop new methods to display ionic currents in computational models of neurons. These tools use color to capture changes in ionic currents and provide insights into how the opening and closing of ion channels shape electrical signals.

The methods developed by Alonso and Marder could be adapted to display the behavior of biochemical reactions or other topics in biology and may, therefore, be useful to analyze data generated by computational models of many different types of cells. Additionally, these methods may potentially be used as educational tools to illustrate the coordinated opening and closing of ion channels in neurons and other fundamental principles of neuroscience that are otherwise hard to demonstrate.

DOI: https://doi.org/10.7554/eLife.42722.002

apply evolutionary techniques to optimize a different family of landscape functions that rely on thresholds or Poincaré sections to characterize the models' solutions.

In some respects, biological systems are a black-box because one cannot read out the values over time of all their underlying components. In contrast, computational models allow us to inspect how all the components interact and this can be used to develop intuitions and predictions about how these systems will respond to perturbations. Despite this, much modeling work focuses on the variables of the models that are routinely measured in experiments, such as the membrane potential. While in the models we have access to all state variables, this information can be hard to represent when many conductances are at play. Similarly, the effect of perturbations – such as the effect of partially or completely blocking or removing a particular channel – can be complex and also hard to display in a compact fashion. Here, we address these difficulties and illustrate two novel visualization methods. We represent the currents in a model neuron using stacked area plots: at each time step, we display the shared contribution of each current to the total current through the membrane. This representation is useful to visualize which currents are most important at each instant and allows the development of insight into how these currents behave when the system is perturbed. Perturbation typically results in drastic changes of the waveform of the activity and these changes depend on the kind of perturbation under consideration. Additionally, we developed a novel representation that relies on computing the probability of $V(t)$, which allows a visualization of these changes. We illustrate the utility of these procedures using models of single neuron bursters or oscillators.

## Results

### Finding parameters: landscape optimization

The numerical exploration of conductance-based models of neurons is a commonplace approach to address fundamental questions in neuroscience (*Dayan and Abbott, 2001*). These models can display much of the phenomenology exhibited by intracellular recordings of single neurons and have the major advantage that many of their parameters correspond to measurable quantities (*Herz et al., 2006*). However, finding parameters for these models so that their solutions resemble experimental observations is a difficult task. This difficulty arises because the models are nonlinear,

they have many state variables and they contain a large number of parameters (*Bhalla and Bower, 1993*). These models are complex, and we are not aware of a general procedure that would allow the prediction of how an arbitrary perturbation in any of the parameters will affect their solutions. The problem of finding sets of parameters so that a nonlinear system will display a target behavior is ubiquitous in the natural sciences. A general approach to this problem consists of optimizing a score function that compares features of the models' solutions to a set of target features. Consequently, landscape-based optimization techniques for finding parameters in compartmental models of neurons have been proposed before (*Achard and De Schutter, 2006*; *Druckmann et al., 2007*; *Ben-Shalom et al., 2012*). Here, we employ these ideas to develop a family of score functions that are useful to find parameters so that their activities reach a desired target.

In this work, we started with a well-studied model of neural activity described previously (*Liu et al., 1998*; *Goldman et al., 2001*; *Prinz et al., 2004*; *O'Leary et al., 2014*). The neuron is modeled according to the Hodgkin-Huxley formalism using a single compartment with eight currents. Following *Liu et al. (1998)*, the neuron has a sodium current, $I_{Na}$; transient and slow calcium currents, $I_{CaT}$ and $I_{CaS}$; a transient potassium current, $I_A$; a calcium-dependent potassium current, $I_{KCa}$; a delayed rectifier potassium current, $I_{Kd}$; a hyperpolarization-activated inward current, $I_H$; and a leak current $I_{leak}$.

We explored the space of solutions of the model using landscape optimization. The procedure consists of three steps. First, we generate voltage traces by integration of *Equation 5* (Materials and methods). We then score the traces using an objective or landscape function that defines a target activity. Finally, we attempt to find minima of the objective function. The procedures used to build objective functions whose minima correspond to sets of conductances that yield the target activities are shown in *Figure 1*. Voltage traces were generated by integration of *Equation 5* and were then scored according to a set of simple measures. The procedure is efficient in part because we chose measures that require little computing power and yet are sufficient to build successful target functions. For example, we avoid the use of Spike Density Functions (SDF) and Fourier transforms when estimating burst frequencies and burst durations. In this section, we describe target

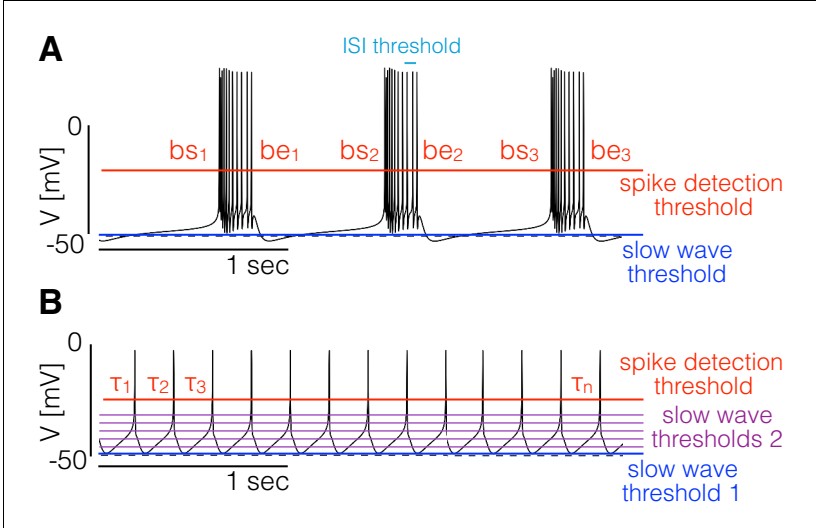

**Figure 1.** Landscape optimization can be used to find models with specific sets of features. (**A**) Example model bursting neuron. The activity is described by the burst frequency and the burst duration in units of the period (duty cycle). The spikes detection threshold (red line) is used to determine the spike times. The ISI threshold (cyan) is used to determine which spikes are bursts starts (bs) and bursts ends (be). The slow wave threshold (blue line) is used to ensure that slow wave activity is separated from spiking activity. (**B**) Example model spiking neuron. We use thresholds as before to measure the frequency and the duty cycle of the cell. The additional slow wave thresholds (purple) are used to control the waveform during spike repolarization.
DOI: https://doi.org/10.7554/eLife.42722.003

functions whose minima correspond to bursting and tonic activity in single compartment models. This approach can also be applied to the case of small circuits of neurons (*Prinz et al., 2004*).

We begin with the case of bursters (*Figure 1A*). We targeted this type of activity by measuring the bursting frequency, the duty cycle, and the number of crossings at a threshold value to ensure that spiking activity is well separated from slow wave activity. To measure the burst frequency and duty cycle of a solution, we first compute the time stamps at which the cell spikes. Given the sequence of values $V = \{V_n\}$ we determine that a spike occurs every time that $V$ crosses the spike detection threshold $T_{sp} = -20mV$ (red in *Figure 1*). We build a sequence of spike times $S = \{s_i\}$ by going through the sequence of voltages $\{V_n\}$ and keeping the values of $n$ for which $V_n \leq T_{sp}$ and $V_{n+1} > T_{sp}$ (we consider upward crossings). Each element $s_i$ of the sequence $S$ contains the time step at which the i-th spike is detected. Bursts are determined from the sequence of spike times $S$; if two spikes happen within a temporal interval shorter than $\delta_{spt} = 100msec$ they are part of a burst. Using this criterion we can find which of the spike times in $S$ correspond to the start and end of bursts. The starts (bs) and ends (be) of bursts are used to estimate the duty cycle and burst frequency. We loop over the sequence of spike times and determine that a burst starts at $s_i$ if $s_{i+1} - s_i < \delta_{spt}$ and $s_i - s_{i-1} > \delta_{spt}$. After a burst starts, we define the end of the burst at $s_k$ if $s_{k+1} - s_k > \delta_{spt}$ and $s_k - s_{k-1} < \delta_{spt}$. When a burst ends we can measure the burst duration as $\delta_b = s_k - s_i$ and since the next burst starts (by definition) at $s_{k+1}$ we also can measure the 'period' (if periodic) of the oscillation as $\tau_b = \delta_b + (s_{k+1} - s_k)$. Every time a burst starts and ends we get an instance of the burst frequency $f_b = \frac{1}{\tau_b}$ and the duty cycle $d_c = \frac{\delta_b}{\tau_b}$. We build distributions of these quantities by looping over the sequence $S$ and define the burst frequency and duty cycle as the mean values $<f_b>$ and $<dc>$. Finally, we count downward crossings in the sequence $V_n$ with *two* slow wave thresholds $\#_{sw}$ (with $t_{sw} = -50 \pm 1mV$) and the total number of bursts $\#_b$ in $S$.

For any given set of conductances, we simulated the model for 20 s and dropped the first 10 s to mitigate the effects of transient activity. We then computed the burst frequency $<f_b>$, the duty cycle $<dc>$, the number of crossings with the slow wave thresholds $\#_{sw}$ and the number of bursts $\#_b$. We discard unstable solutions; a solution is discarded if $std(\{f_b\}) \geq (<f_b> \times 0.1)$ or $std(\{dc\}) \geq (<dc> \times 0.2)$. If a solution is not discarded, we can use the following quantities to measure how close it is to the target behavior,

$$
\begin{aligned}
E_f &= (f_{tg} - <f_b>_i)^2 \\
E_{dc} &= (dc_{tg} - <dc>_i)^2 \\
E_{sw} &= (\frac{\#_{sw}}{2} - \#_b)^2
\end{aligned}
\tag{1}
$$

Here, $E_f$ measures the mismatch of the bursting frequency of the model cell with a target frequency $f_{tg}$ and $E_{dc}$ accounts for the duty cycle. $E_{sw}$ measures the difference between the number of bursts and the number of crossings with the slow wave thresholds $t_{sw} = -50 \pm 1mV$. Because we want a clear separation between slow wave activity and spiking activity, we ask that $\#_{sw} = \#_b$. Note that if during a burst $V$ goes below $t_{sw}$ this solution would be penalized (factor $\frac{1}{2}$ accounts for using two slow wave thresholds). Let g denote a set of parameters, we can then define an objective function

$$
E(\mathbf{g}) = \alpha E_f + \beta E_{dc} + \gamma E_{sw},
\tag{2}
$$

where the weights $(\alpha, \beta, \gamma)$ determine the relative importance of the different sources of penalties. In this work we used $\alpha = 1$, $\beta = 100$, $\gamma = 1$, and the penalties $E_i$ were calculated using $T = 10$ seconds with $dt = 0.1$ msecs. The target behavior for bursters was defined by $dc_{tg} = 0.2$ (duty cycle 20%) ($dc_{tg} = 0.2$) and bursting frequency $f_{tg} = 1Hz$.

We can use similar procedures to target tonic spiking activity. Note that the procedure we described previously to determine bursts from the sequence of spike times $S$ is also useful in this case. If a given spike satisfies the definition of burst start and it also satisfies the definition of burst end then it is a single spike and the burst duration is zero. Therefore, we compute the bursts and duty cycles as before and ask that the the target duty cycle is zero.

There are multiple ways to produce tonic spiking in this model and some solutions display very different slow wave activity. To further restrict the models, we placed a middle threshold at $t_{mid} = -35mV$ and detected downward crossings at this value. We defined $E_{lag}$ as the lag between

the upward crossings at the spiking threshold ($t_{spk} = -20mV$) and downward crossings at $t_{mid}$. $E_{lag}$ is useful because it takes different values for tonic spikers than it does for single-spike bursters even though their spiking patterns can be identical. Finally, we found that the model attempts to minimize $E_{lag}$ at the expense of hyperpolarizing the membrane beyond $-50mV$ and introducing a wiggle that can be different in different solutions. To penalize this we included additional thresholds between $-35mV$ and $-45mV$, counted the number of downward crossings at these values $\#_{mid_i}$, and asked that these numbers are equal to the number of spikes $\#_s$. With these definitions, we define the partial errors as before,

$$E_f = (f_{tg} - <f_b>_i)^2$$
$$E_{dc} = (dc_{tg} - <dc>_i)^2$$
$$E_{mid} = \sum_i (\#_{mid_i} - \#_s)^2 \qquad (3)$$
$$E_{sw} = (\#_{sw})^2.$$

The total error as a function of the conductances reads as follows,

$$E(\mathbf{g}) = \alpha E_f + \beta E_{dc} + \gamma E_{mid} + \delta E_{sw} + \eta E_{lag}. \qquad (4)$$

The values $\alpha = 1000$, $\beta = 1000$, $\gamma = 100$, $\delta = 100$ and $\eta = 1$, produce solutions that are almost identical to the one displayed in *Figure 1B*.

In all cases, evaluation of the objective functions requires that the models are simulated for a number of seconds and this is the part of the procedure that requires most computing power. Longer simulations will provide better estimations for the burst frequency and duty cycle of the cells, but will linearly increase the time it takes to evaluate the objective function. If the simulations are shorter, evaluations of the objective function are faster but the minimization may be more difficult due to transient behaviors and its minima may not correspond to stable solutions. In this work, we minimized the objective function using a standard genetic algorithm (*Holland, 1992*; *Goldberg and Holland, 1988*). The choice of the optimization routine and the choice of the numerical scheme for the simulations are independent of the functions. See Materials and methods for details on the how we performed this optimization. The same functions can be utilized to estimate parameters in models with different channel types.

## Visualizing the dynamics of ionic currents: currentscapes

Most modeling work focuses on the variables of the models that are routinely measured in experiments such as the membrane potential as is shown in *Figure 2A* for a bursting neuron. While in the models we have access to all state variables, this information can be hard to represent when several current types are at play. One difficulty is that some currents like $Na$ and $Kd$ vary over several orders of magnitude, while other currents like the *leak* and $H$ span smaller ranges. Additionally, the relative contribution of each current to the total flux through the membrane varies over time. Here, we introduce a novel representation that is simple and permits displaying the dynamics of the currents in a cohesive fashion.

At any given time stamp, we can compute the total inward and outward currents. We can then express the values of each current as a percentage of this quantity. The normalized values of the currents at any time can be displayed as a pie chart representing the share of each current type (*Figure 2B*). Because we want to observe how these percentages change in time, we display the shares in a bar instead of a disk. The currentscapes are constructed by applying this procedure to all time stamps and stacking the bars. These types of plots are known as stacked area plots and their application to this problem is novel. *Figure 2C* shows the currentscape of a periodically bursting model neuron over one cycle. The shares of each current type to the total inward and outward currents are displayed in colors, and the total inward and outward currents are represented by the filled black curves in logarithmic scale in the top and bottom.

## Visualizing changes in the waveforms as a parameter is changed

To visualize changes in the activity as a conductance is gradually removed we computed the distribution of membrane potential $V$ values. This reduction contains information about the waveform of the

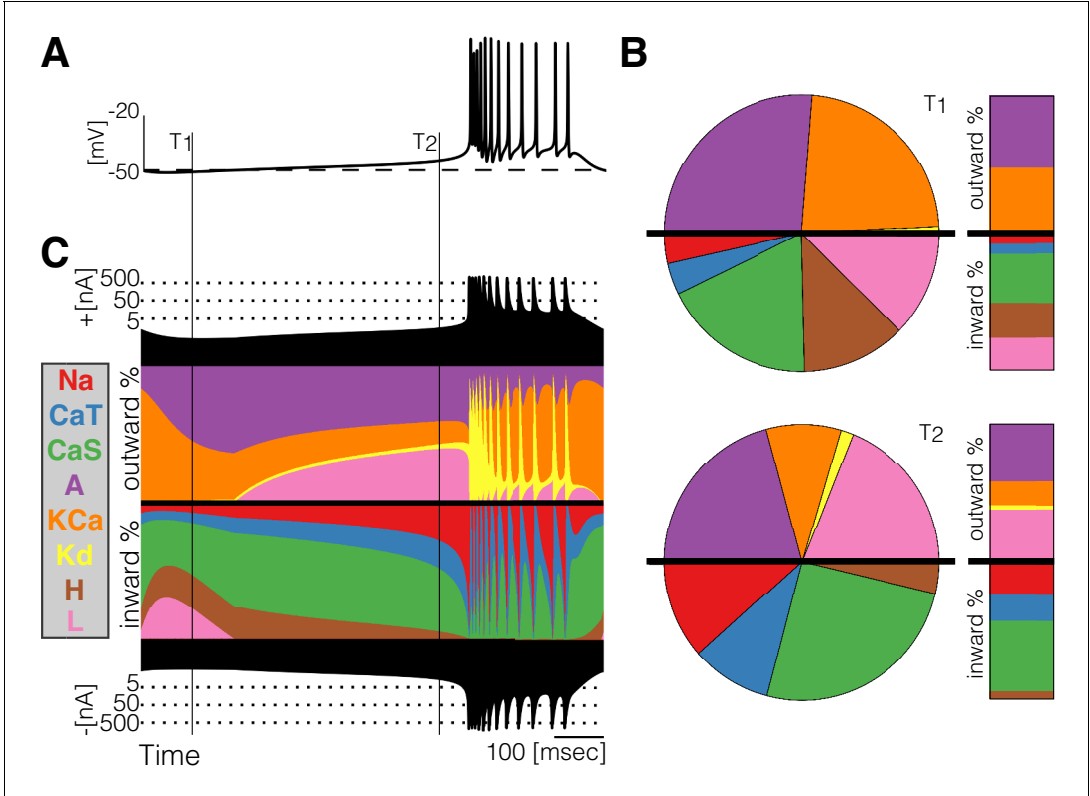

**Figure 2.** Currentscape of a model bursting neuron. A simple visualization of the dynamics of ionic currents in conductance-based model neurons. (**A**) Membrane potential of a periodic burster. (**B**) Percent contribution of each current type to the total inward and outward currents displayed as pie charts and bars at times $T_1$ and $T_2$ (**C**) Percent contribution of each current to the total outward and inward currents at each time stamp. The black filled curves on the top and bottom indicate total inward outward currents respectively on a logarithmic scale. The color curves show the time evolution of each current as a percentage of the total current at that time. For example, at $t = T_1$ the total outward current is $\approx 2.5nA$ and the orange shows a large contribution of $KCa$. At $t = T_2$ the total outward current has increased to $\approx 4nA$ and the $KCa$ current is contributing less to the total.

DOI: https://doi.org/10.7554/eLife.42722.004

membrane potential, while all temporal information such as frequency can no longer be recovered. The number of times that a given value of $V$ is sampled is proportional to the time the system spends at that value. *Figure 3A* shows the distribution of $V$ for a periodic burster with $f_b \approx 1Hz$ and $d_c \approx 20\%$ sampled from 30 s of simulation. The count is larger than $10^4$ for values between $-52mV$ and $-40mV$, and smaller than $10^3$ for $V$ between $-35mv$ and $20mV$. The areas of the shaded regions are proportional to the probability that the system will be observed at the corresponding $V$ range (*Figure 3B*). Note that the area of the dark gray region is $10^5$ while the light gray is $0.5 \times 10^4$, so the probability that the cell is, at any given time, in a hyperpolarized state is more than 20 times larger than the probability that the cell is spiking.

The distribution of $V$ features sharp peaks. In many cases, the peaks in these distributions correspond to features of the waveform, such as the amplitudes of the individual spikes, or the minimum membrane potential (see *Figure 3—figure supplement 1*). This happens because every time the membrane potential reaches a maxima or minima (in time) the derivative $\frac{dV}{dt}$ is close to zero. The system spends more time close to values of $V$ where the velocity $\frac{dV}{dt}$ is small than in regions where $\frac{dV}{dt}$ is large, as it occurs during the flanks of spikes. Therefore, when we sample a solution at a random instant, it is more likely that $V$ corresponds to the peak of a spike than to either flank of the spike, while the most likely outcome is that $V$ is in the hyperpolarized range $(< -40mV)$. In this particular burster, there are 12 spikes in the burst but there are only 7 peaks in the distribution (between $10mV$ and $20mV$); some spikes have similar amplitudes so they add to a larger peak in the distribution. The overall or total amplitude of the oscillation can be read from the distribution since the count of $V$ is zero outside a range $(-52mV$ to $20mV)$. These distributions can also be represented by a graded bar

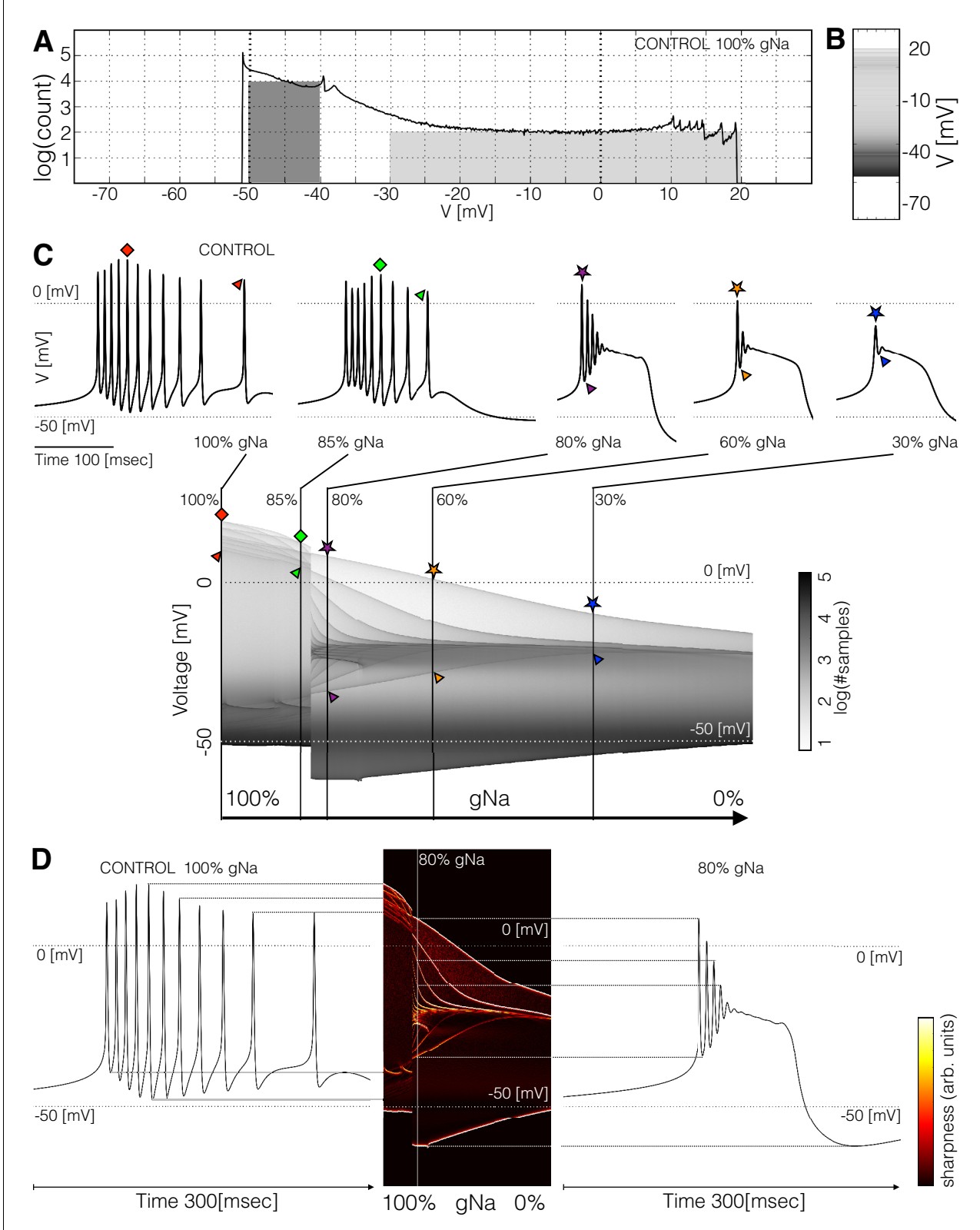

**Figure 3.** Membrane potential $V$ distributions. (A) Distribution of membrane potential $V$ values. The total number of samples is $N = 2.2 \times 10^6$. Y-axis scale is logarithmic. The area of the dark shaded region can be used to estimate of the probability that the activity is sampled between $-50mV$ and $-40mV$, and the area of the light shaded region is proportional to the probability that $V(t)$ is sampled between $-30mV$ and $20mV$. The area of the dark region is 20 times larger than the light region. (B) The same distribution in (A) represented as a graded bar. (C) Distribution of $V$ as a function of $V$ and

*Figure 3 continued on next page*

*Figure 3 continued*

*gNa*, and waveforms for several *gNa* values.The symbols indicate features of the waveforms and their correspondence to the ridges of the distribution of *V*. (D) Waveforms under two conditions and their correspondence to the ridges of the distribution of *V*. The ridges were enhanced by computing the derivative of the distribution along the *V* direction.

DOI: https://doi.org/10.7554/eLife.42722.005

The following figure supplement is available for figure 3:

**Figure supplement 1.** Probability distributions of membrane potential.

DOI: https://doi.org/10.7554/eLife.42722.006

as shown in *Figure 3B*. As conductances are gradually removed the waveform of the activity changes and so does the distribution of *V* values.

*Figure 3C* shows how the distribution of *V* changes as *gNa* is decreased. The waveforms at a few values of *gNa* are shown for reference. For each value in the range ($100\%gNa$ to $0\%gNa$ with $N = 1001$ values) we computed the count $p(V, gNa)$ and display $log_{10}(p(V, gNa) + 1)$ in gray scales. In this example, the cell remains in a bursting regime up to $\approx 85\%gNa$ and transitions abruptly into a single-spike bursting mode for further decrements ($\%80gNa$). The spikes produce thin ridges in the distribution that show how their individual amplitudes change. The colored symbols indicate the correspondence between features in the waveform and ridges in the distribution. In this example, the peak amplitudes of the spikes are similar for values of *gNa* greater than $\%85gNa$. After the transition, the amplitudes of the spikes are very different; two spikes go beyond $0mV$ and the rest accumulate near $-25mV$. As $gNa \rightarrow 0$ the oscillations collapse onto a small band at $\approx -20mV$ and only one spike is left.

The distributions allow the visualization of the amplitudes of the individual spikes, the slow waves, and other features as the parameter *gNa* is changed. To highlight ridges in the distributions, the center panel in *Figure 3D* shows the derivative $\partial_V log_{10}(p(V))$ in color. This operation is similar to performing a Sobel filtering (*Sobel and Feldman, 1968*) of the image in *Figure 3C*. The traces on each side of this panel correspond to the control (left) and $80\%gNa$ conditions. Notice how the amplitudes of each spike, features of the slow wave, and overall amplitude correspond to features in the probability distribution. This representation permits displaying how the features of the waveform change for many values of the perturbation parameter *gNa*.

## The maximal conductances do not fully predict the currentscapes

We explored the solutions of a classic conductance-based model of neural activity using landscape optimization and found many sets of parameters that produce similar bursting activity. Inspired by intracellular recording performed in the Pyloric Dilator (*PD*) neurons in crabs and lobsters we targeted bursters with frequencies $f_b \approx 1Hz$ and duty cycles $dc \approx 20\%$. We built 1000 bursting model neurons and visually inspected the dynamics of their currents using their currentscapes. Based on this, we selected six models that display similar membrane activity via different current compositions for further study. Because the models are nonlinear, the relationship between the dynamics of a given current type and the value of its maximal conductance is non-trivial. *Figure 4* shows the values of the maximal conductances in the models (top) and their corresponding activity together with their currentscapes (bottom).

It can be difficult to predict the currentscapes based on the values of the maximal conductances. In most cases, it appears that the larger the value of the maximal conductance, the larger the contribution of the corresponding current. However, this does not hold in all cases. For example, burster (f) shows the largest *A* current contribution, but bursters (c) and (e) have larger values of *gA*. The maximal conductance of the *CaS* current is low in model (f) but the contribution of this current to the total is similar to that in models (a) and (b). The values of *gKCa* are similar for bursters (e) and (f) but the contribution of this current is visibly different in each model.

## Response to current injection

The models produce similar activity with different current dynamics. To further reveal differences in how these activities are generated, we subjected the models to simple perturbations. We begin describing the response to constant current injections in *Figure 5*. *Figure 5A* and *Figure 5B* show

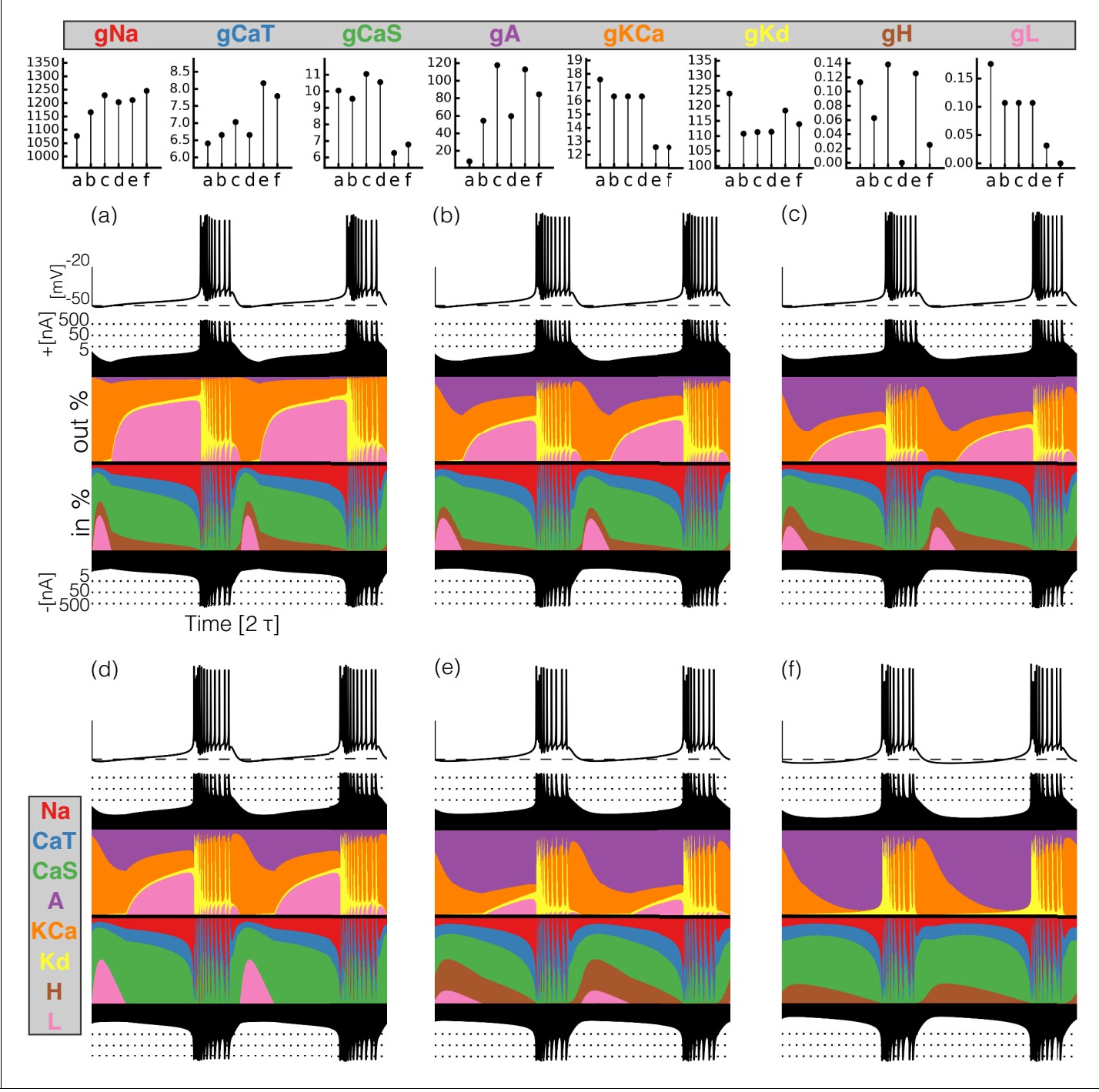

**Figure 4.** Currentscapes of model bursting neurons. (top) Maximal conductances of all model bursters. (bottom) The panels show the membrane potential of the cell and the percent contribution of each current over two cycles.

DOI: https://doi.org/10.7554/eLife.42722.007

the membrane potential of model (a) for different values of injected current. In control, the activity corresponds to regular bursting and larger depolarizing currents result in a plethora of different regimes. The distributions of inter-spike intervals (ISI) provide a means to characterize these regimes (**Figure 5C**). When the cell is bursting regularly such as in control and in the $0.8nA$ condition, the interspike interval distributions consist of one large value that corresponds to the interburst interval

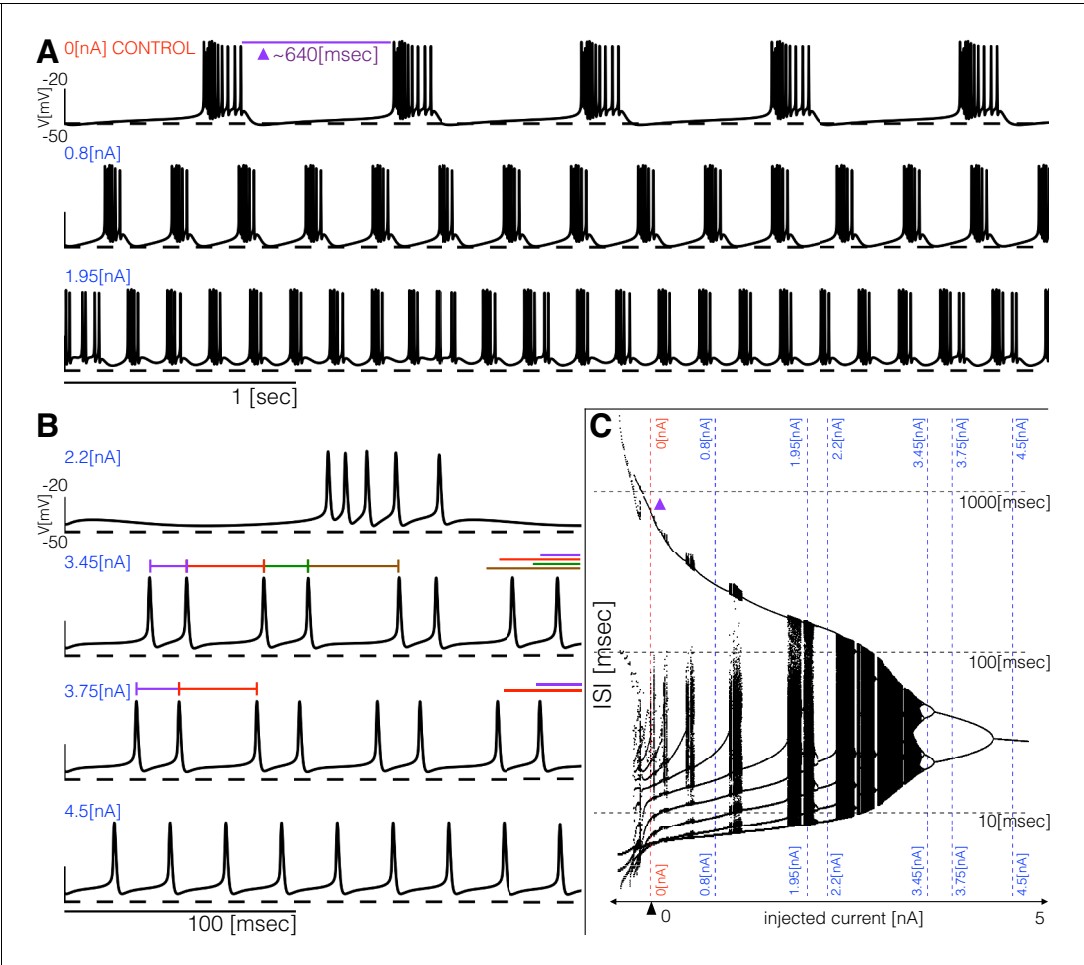

**Figure 5.** Response to current injections and interspike-intervals (ISI) distributions of model (a). (**A**) (top) Control traces (no current injected $0nA$), regular bursting ($0.8nA$), irregular bursting $1.95nA$. (**B**) (top) Fast regular bursting ($f_b \approx 6Hz$), quadruplets ($3.45nA$), doublets ($3.75nA$) and singlets ($4.5nA$) (tonic spiking). (**C**) ISI distributions over a range of injected current.
DOI: https://doi.org/10.7554/eLife.42722.008

($\approx 640msec$ in control) and several smaller values around $10msec$ which correspond to the ISI within a burst. There are values of current for which the activity appears irregular and correspondingly, the ISI values are more diverse. **Figure 5B** shows the response of the model to larger depolarizing currents. The activity undergoes a sequence of interesting transitions that result in tonic spiking. When $I_e = 3.45nA$ the activity is periodic and there are 4 ISI values. Larger currents result in 2 ISI values and tonic spiking produces one ISI value. **Figure 5C** shows the ISI distributions (y-axis, logarithmic scale) for each value of injected current (x-axis).

All these bursters transition into tonic spiking regimes for depolarizing currents larger than $5nA$ but they do so in different ways. To explore these transitions in detail, we computed the inter-spike interval (ISI) distributions over intervals of $60sec$ for different values of the injected current. **Figure 6** shows the ISI distributions for the six models at $N = 1001$ equally spaced values of injected current over the shown range. The y-axis shows the values of all ISIs on a logarithmic scale and the x-axis corresponds to injected current. In the control, the ISI distribution consists of a few small values ($<100msec$) that correspond to the ISIs of spikes within a burst, and a single larger value ($>100msec$) that corresponds to the interval between the last spike of a burst and the first spike of the next burst. When the cell fires tonically the ISI distributions consist of a single value. The ISI distributions exhibit complicated dependences on the control parameter that result in beautiful patterns. For some current values, the cells produce small sets of ISI values indicating that the activity is periodic. However, this activity is quite different across regions. Interspersed with the regions of periodicity

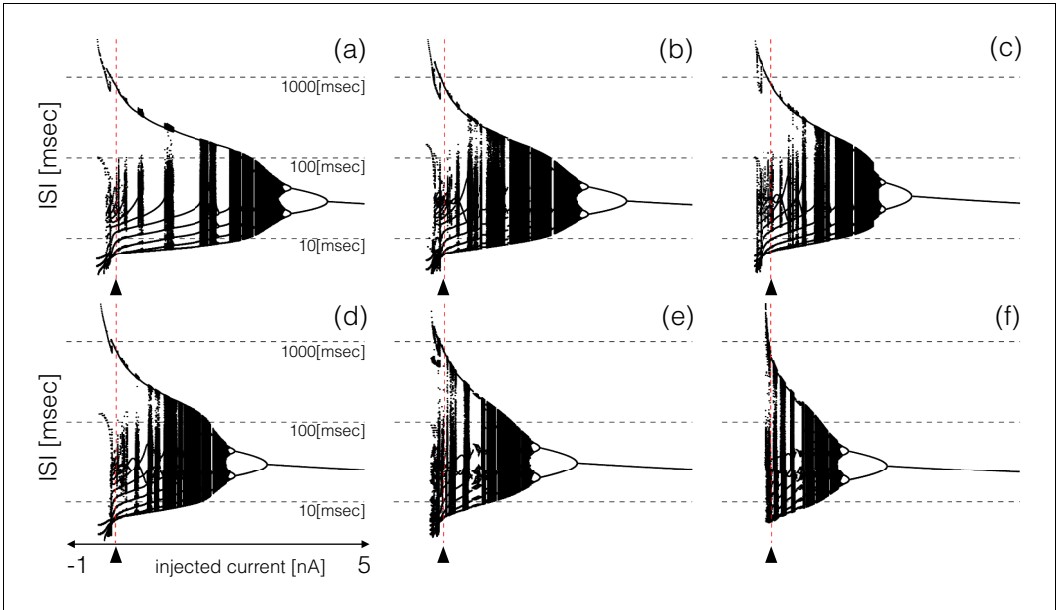

**Figure 6.** ISI distributions of the six model bursting neurons over a range of injected current. The panels show all ISI values of each model burster over a range on injected currents (vertical axis is logarithmic). All bursters transition into tonic spiking regimes for injected currents larger than $5nA$ and the details of the transitions are different across models.

DOI: https://doi.org/10.7554/eLife.42722.009

there are regions where the ISI distributions densely cover a band of values indicating non-periodic activity. Overall the patterns feature nested forking structures that are reminiscent of classical period doubling routes to chaos (*Feigenbaum, 1978*; *Canavier et al., 1990*).

## Extracting insights from these visualization tools

Detailed conductance-based models show complex and rich behaviors in response to all kinds of perturbations. There is a vast amount of information that can be seen in these models and their visualizations in *Figures 7 - 15*. It is entirely impossible for us to point out even a fraction of what can be seen or learned from these figures. Nonetheless, we will illustrate a few examples of what can be seen using these methods, knowing that these details will be different for models that are constructed in the future and analyzed using these and similar methods.

## Perturbing the models with gradual decrements of the maximal conductances

*Figures 7* and *8* show the effects of gradually decreasing each of the currents in these bursters from $100\%$ to $0\%$ for all six models. This type of analysis might be relevant to some kinds of pharmacological manipulations or studies of neuromodulators that decrease a given current. The figures show 3 s of data for each condition. In all panels, the top traces correspond to the control condition ($100\%$) and the traces below show the activity that results from decreasing the maximal conductance. The dashed lines are placed for reference at $-50mV$ and $0mV$. Each panel shows the traces for 11 values of the corresponding maximal conductance equally spaced between $100\%$ (control) and $0\%$ (completely removed). Each row of panels corresponds to a current type and the columns correspond to the different model bursters. *Figure 7* displays the perturbations for the inward currents and *Figure 8* shows the outward and leak currents.

Taken together *Figures 7* and *8* illustrate that each model (a-f) changes its behavior differently in response to decreases in each current. Additionally, decreases in some currents have only relatively small effects but decreases in others have much more profound effects. Because the description of all that can be seen in these figures is beyond the scope of this paper, we chose to focus on the effects of decreasing the $CaT$ because it has rich and unexpected behaviors.

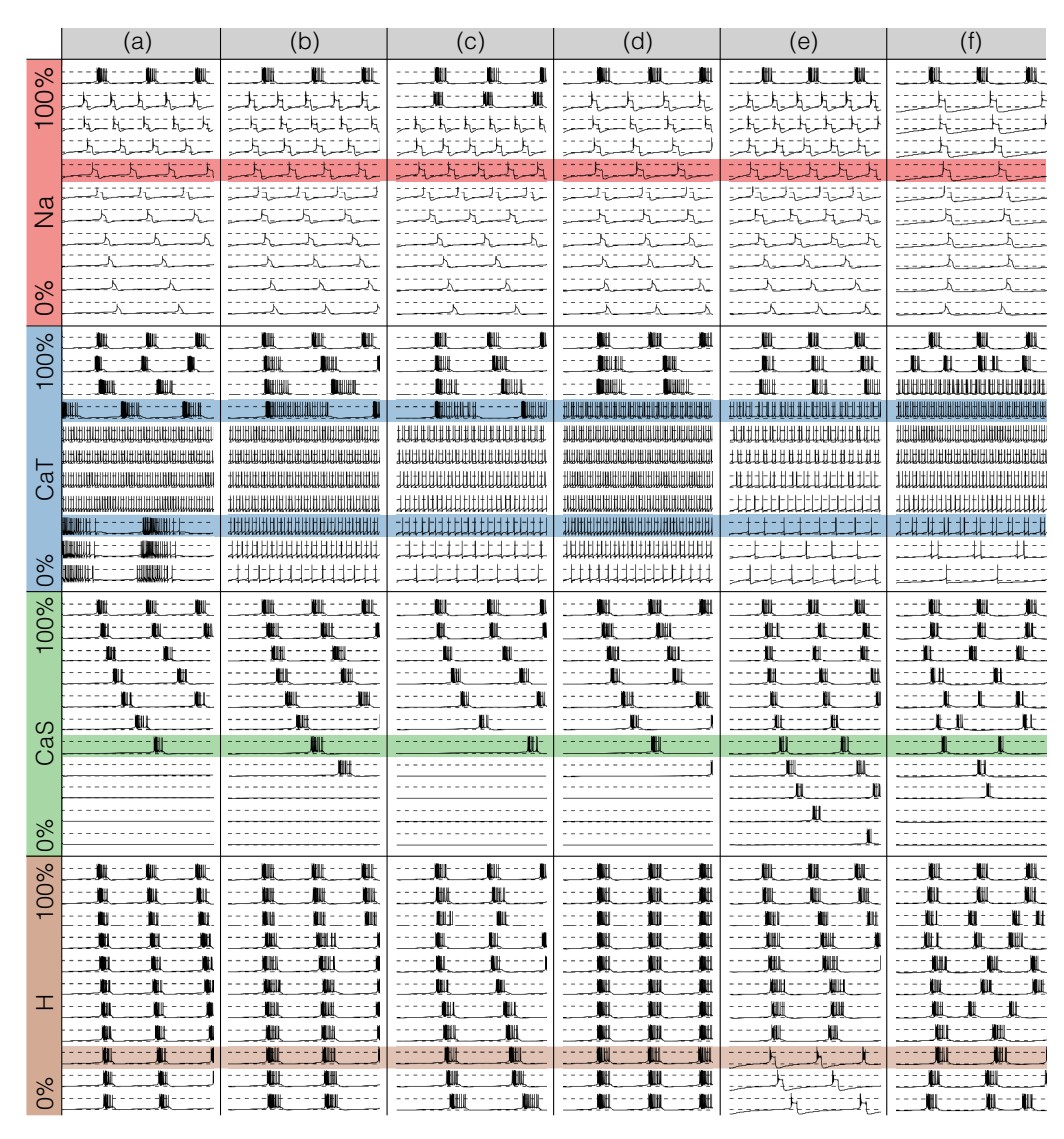

**Figure 7.** Effects of decreasing maximal conductances: inward currents. The figure shows the membrane potential $V$ of all model cells as the maximal conductance $g_i$ of each current is gradually decreased from $100\%$ to $0\%$ . Each panel shows 11 traces with a duration of 3 s. Dashed lines are placed at $0mV$ and $-50mV$. The shading indicates values of maximal conductance for which the activity the models differs the most.

DOI: https://doi.org/10.7554/eLife.42722.010

The effect of decreasing the $CaT$ conductance is quite diverse across models. The activities of the models at the intermediate values of $gCaT$ shows visible differences. When $gCaT \rightarrow 0.7gCaT$ models (a), (b) and (c) show bursting activity at different frequencies and with different duty cycles. Models (d), (e) and (f) become tonic spikers at this condition, but their frequencies are different. Note that in the case of model (e) the spiking activity is not regular and the ISIs take several different values. When $gCaT \rightarrow 0.2gCaT$ most models spike tonically but now (e) is regular and (f) shows doublets. When $CaT$ is completely removed, most models transition into a tonic spiking regime with the exception of model (a), that displays a low frequency bursting regime with duty cycle $\approx 0.5$.

## Gradually removing one current impacts the dynamics of all currents

Decreasing any conductance can trigger qualitative changes in the waveform of the membrane potential and in the contributions of each current to the activity. In *Figure 9* we plot currentscapes for the effects of decreasing $CaT$ in model (f). This allows us to examine at higher resolution the

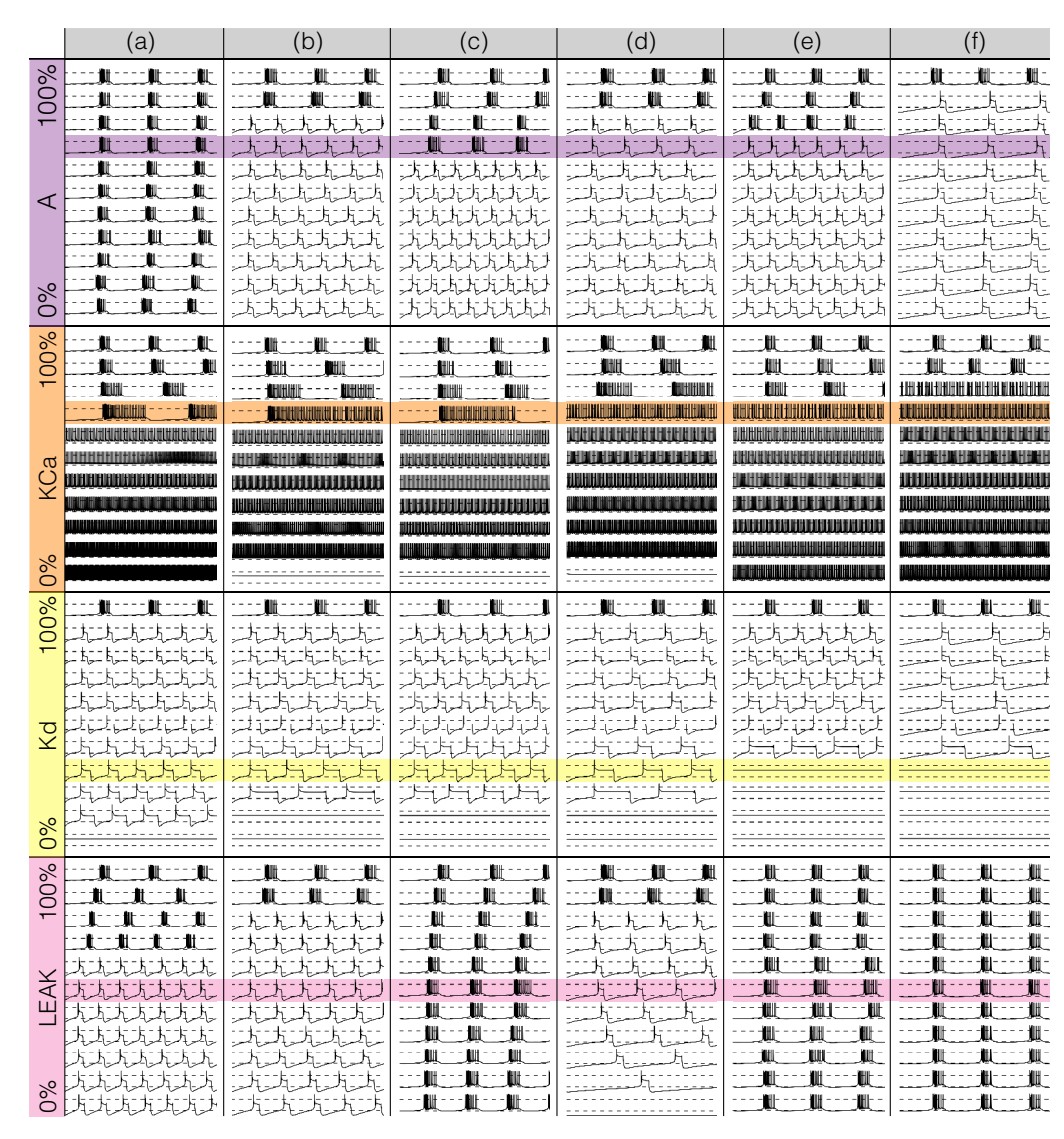

**Figure 8.** Effects of decreasing maximal conductances: outward currents. The figure shows the membrane potential $V$ of all model cells as the maximal conductance $g_i$ of each current is gradually decreased from 100% to 0%. Each panel shows 11 traces with a duration of 3 s. Dashed lines are placed at $0 mV$ and $-50 mV$. The shading indicates values of maximal conductance for which the activity the models differs the most.

DOI: https://doi.org/10.7554/eLife.42722.011

changed contributions of currents that give rise to the interesting dynamics seen in *Figure 7*. Each panel in *Figure 9* corresponds to a different decrement value and shows the membrane potential on top, and the currentscapes at the bottom. The top panels show 1 second of data and correspond to the $100\% gCaT$ (control), $90\% gCaT$ and $80\% gCaT$ conditions. The center panels show 0.1 s of data for decrements ranging from 70% to 20% and the bottom panels show 2 s for the 10% and 0% conditions. As $CaT$ is gradually removed the activity transitions from a bursting regime to a tonic spiking regime.

When $gCaT \rightarrow 90\% gCaT$ the neuron produces bursts but these become irregular and their durations change. Decreasing the conductance to $80\% gCaT$ results in completely different activity. The spiking pattern appears to be periodic but there are at least three different ISI values. It is hard to see changes in the $CaT$ contribution across these conditions, but changes in other currents are more discernible. The contribution of the $A$ current that is large in the control and $90\% gCaT$ conditions, is much smaller in the $80\% gCaT$ condition. Additionally, the $Na$ and $KCa$ currents show larger

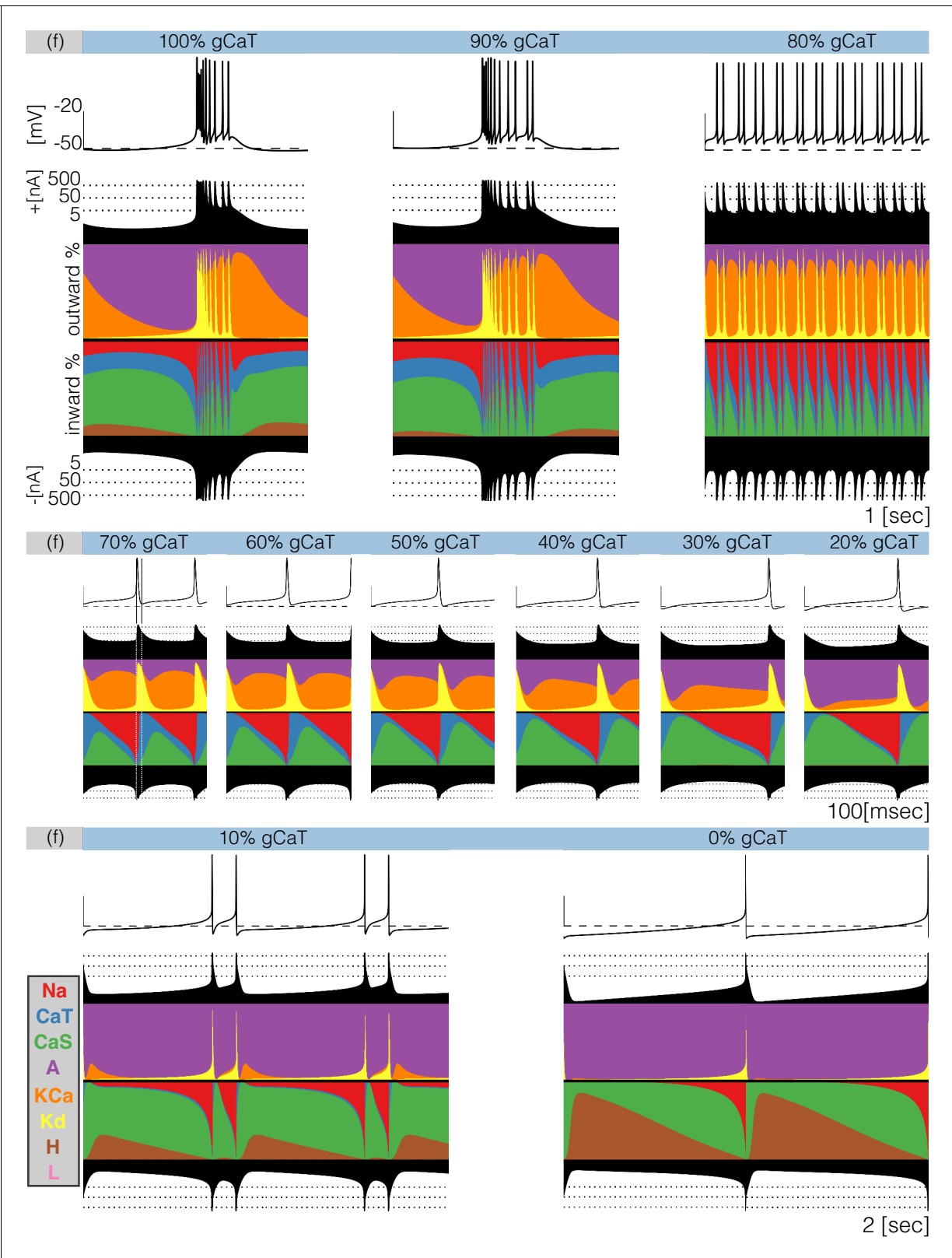

**Figure 9.** Decreasing $CaT$ in model (f). The figure shows the traces and the currentscapes of model (f) as $CaT$ is gradually decreased. Top panels show 1 second of data, center panels show 0.1 seconds and the bottom panels show 2 seconds (see full traces in *Figure 8*).

DOI: https://doi.org/10.7554/eLife.42722.012

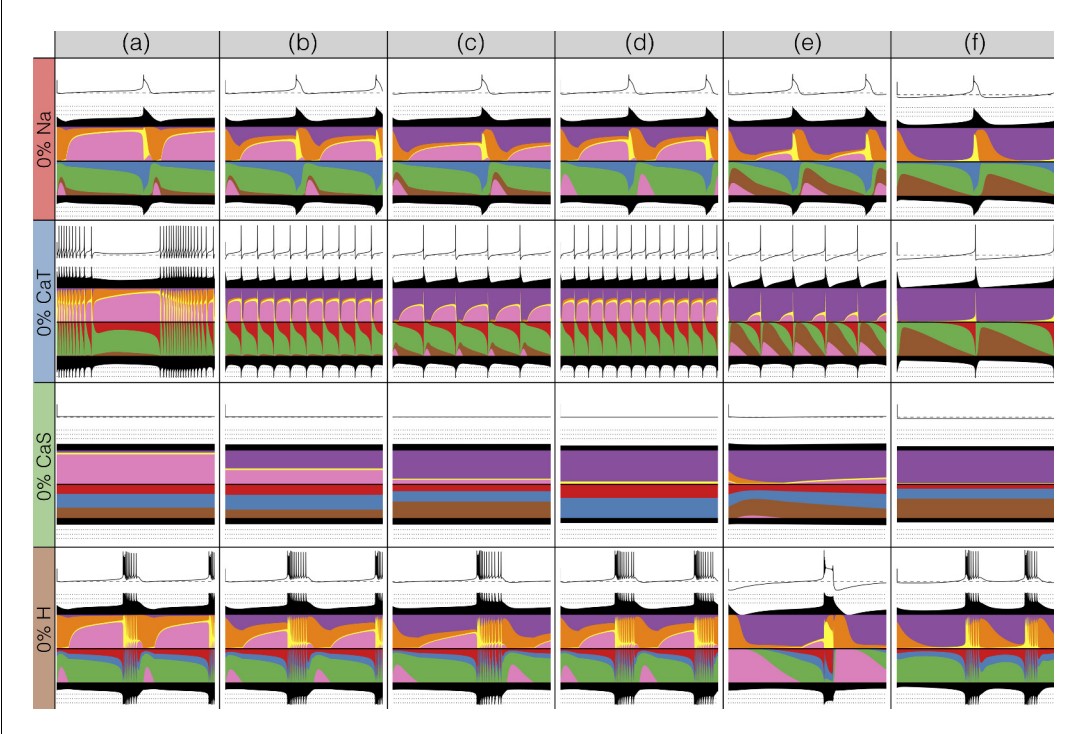

**Figure 10.** Complete removal of one current: inward currents. The figure shows the traces and currentscapes for all bursters when one current is completely removed.

DOI: https://doi.org/10.7554/eLife.42722.013

contributions, the $CaS$ current contributes less and the $H$ current is negligible. Further increments in simulated blocker concentration result in tonic spiking regimes with frequencies ranging from $\approx 20Hz$ to $\approx 10Hz$. The center panels in *Figure 9* show the currentscapes for these conditions on a different time scale to highlight the contributions of $CaT$. The leftmost panel shows the $70\%gCaT$ condition. In this panel, we placed vertical lines indicating the time stamps at which the peak of the spike and the minimum occur. Notice the large contribution of the $Na$ current prior to the peak of the spike, and the large contribution of the $Kd$ current for the next $\approx 10msec$. When the membrane potential is at its minimum value the $CaT$ current dominates the inward currents and remains the largest contributor for the next $\approx 10msec$. The $CaT$ current reduces its share drastically by the time the $Na$ current is visible and $CaS$ takes over. The contribution of $CaT$ remains approximately constant during repolarization and vanishes as the membrane becomes depolarized and the $Na$ current becomes dominant. The effect of removing $CaT$ is visible on this scale. The waveform of the contribution remains qualitatively the same: largest at the minimum voltage and approximately constant until the next spike. However, the contribution of $CaT$ during repolarization becomes smaller, and for larger conductance decrements results in a thinner band. Finally, the bottom panels show the cases $10\%gCaT$ and $0\%gCaT$ which correspond to a two-spike burster and a tonic spiker, respectively. Note that even though the contribution of $CaT$ is barely visible, complete removal of this current results in a very different pattern. The activity switched from bursting to spiking and the current composition is different; $KCa$ disappeared in the $0\%$ condition and the $A$ current takes over. Notice also the larger contribution of the $H$ current.

## Modeling current deletions

There has been a great deal of work studying the effects of genetic and/or pharmacological deletions of currents. One of the puzzles is why some currents, known to be physiologically important, can have relatively little phenotype in some, or all individuals. For this reason in *Figures 10* and *11*, we show the effects of deletion of each current in all six models. Each panel shows 2 seconds of

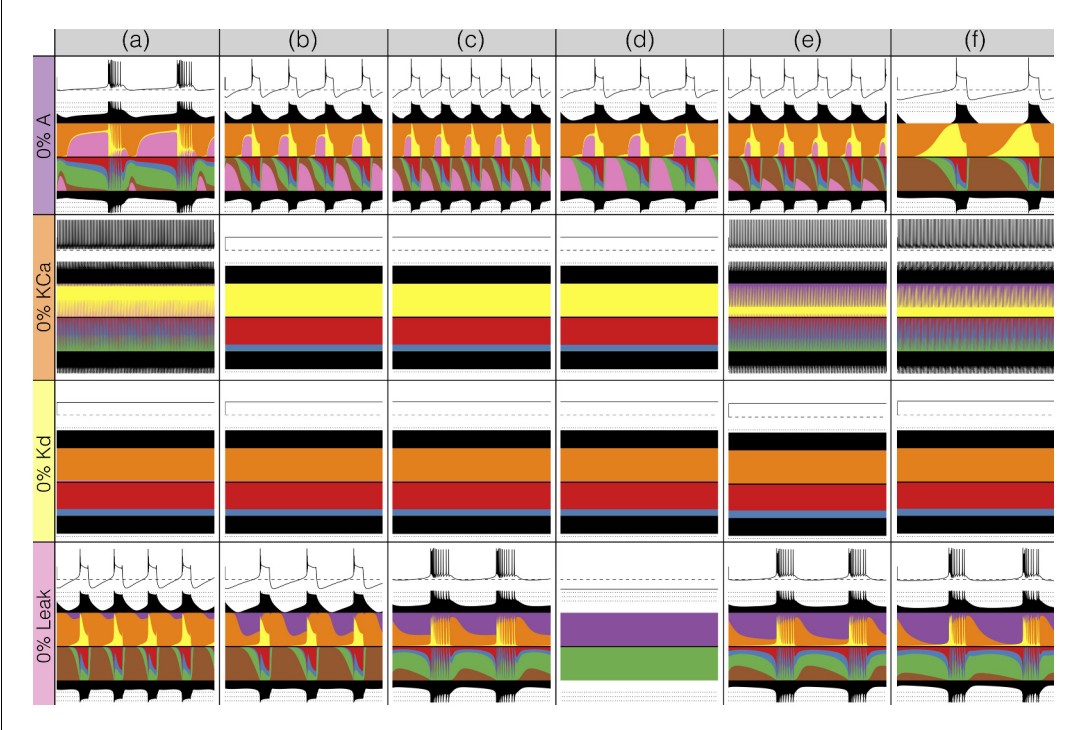

**Figure 11.** Complete removal of one current: outward currents. The figure shows the traces and currentscapes for all bursters when one current is completely removed.

DOI: https://doi.org/10.7554/eLife.42722.014

data. The inward currents are portrayed in *Figure 10* and the outward and leak currents are shown in *Figure 11*.

Removal of some currents has little obvious phenotype differences across the population although the currentscapes are different, such as seen for the $gNa$ and $gCaS$ cases. Removal of some currents produces similar phenotypes in most, but not all of the six models as seen in the $gH$ and $gA$ cases. Removal of $Kd$ had virtually identical effects both on the phenotype and the currents. For other currents, such as $KCa$ and the *Leak*, we find two types of responses with nearly half of the models for each case (the exception is model (d) *Leak*). In the case of the $CaT$ current both the phenotype and the currents composition are very diverse across models.

## Changes in waveform as conductances are gradually decreased

A fuller description of the behavior/phenotype of all of the models for all values of conductance decrements can be seen in *Figures 12* and *13*. These figures use the probability scheme described in *Figure 3* and *Figure 3—figure supplement 1*. Using these methods, it is possible to see exactly how the waveforms change and the boundaries of activity for each model and each conductance. The panels show the ridges of the probability distributions $p(V)$ of the membrane potential $V(t)$ for 1001 values of maximal conductance values (see Materials and methods). The probability of $V(t)$ was computed using 30 s of data after dropping a transient period of 120 s. It was estimated using $N_b = 1001$ bins in the range $(-70, 35) mv$ and $N \approx 2 \times 10^6$ samples for each maximal conductance value. The system spends more time in regions where $\frac{dV}{dt} \approx 0$ and is sampled more at those values. Therefore, features such as the amplitudes of the spikes appear as sharp peaks in the probability distributions. To highlight these peaks and visualize how they change as currents are gradually decreased, we plot the derivative or sharpness of the distribution in colors (see color scale in *Figure 3D*). Overall, these plots show that for any given current, there are ranges of the conductance values where a small change results in a smooth deformation of the waveform, and there are specific values at which abrupt transitions take place. As before there is too much detail to describe

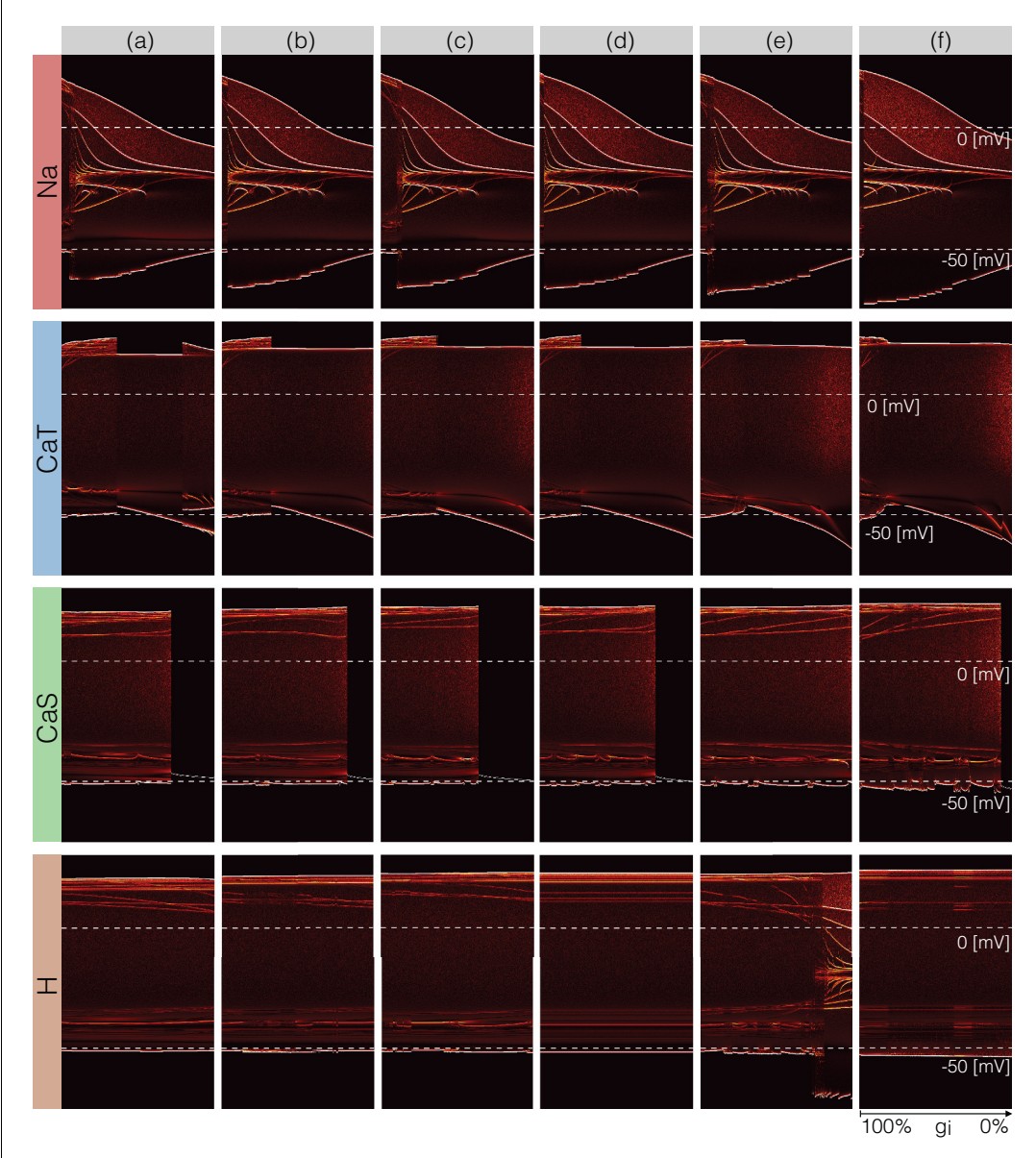

**Figure 12.** Changes in waveform as currents are gradually removed. Inward currents. The figure shows the ridges of the probability distribution of $V(t)$ as a function of $V$ and each maximal conductance $p(V, g_i)$. The ridges of the probability distributions appear as curves and correspond to values of $V$ where the system spends more time, such as extrema. The panels show how different features of the waveform such as total amplitude, and the amplitude of each spike, change as each current is gradually decreased.
DOI: https://doi.org/10.7554/eLife.42722.015

everything in these figures so we will discuss a subset of the features highlighted by this representation.

The top rows in *Figure 12* correspond to removing the $Na$ current in the models. Note that the minimum value of $V$ in control (left) is close to $-50mV$ and a small decrement in $gNa$ results in larger amplitude. The colored curves inside the envelopes correspond to the spikes' amplitudes and features of the slow waves. For instance, when the $Na$ current is completely removed (right) the amplitude of the oscillation is $\approx 40mV$ and the activity corresponds to a single-spike bursting mode. The spike amplitude is given by the top edge of the colored region and the curve near $\approx -20mV$ indicates the burst 'belly': the membrane hyperpolarizes slowly after spike termination and there is a wiggle at this transition.

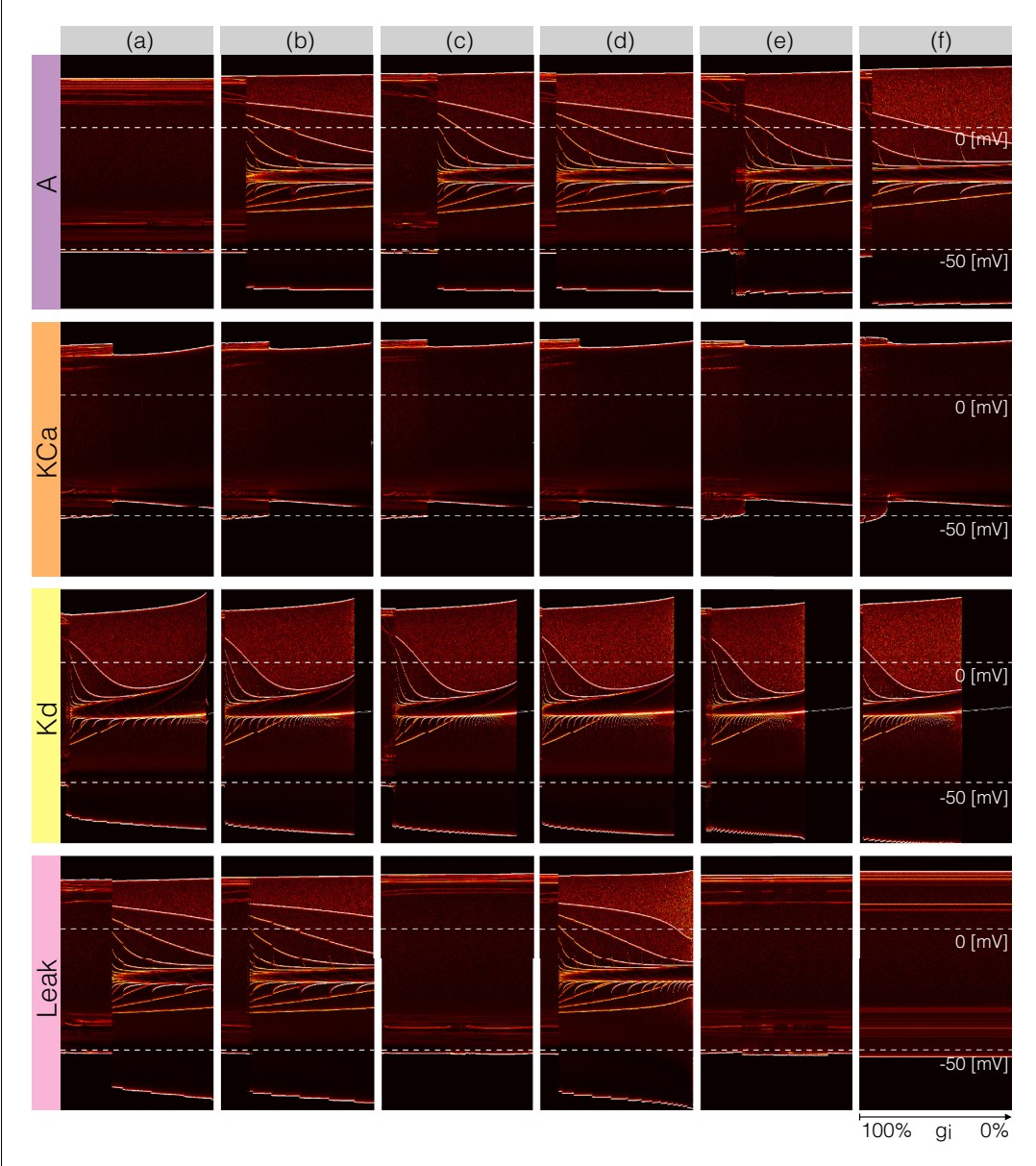

**Figure 13.** Changes in waveform as currents are gradually removed. Outward and leak currents. The figure shows the ridges of the probability distribution of $V(t)$ as a function of $V$ and each maximal conductance $p(V, g_i)$. See **Figure 12**.
DOI: https://doi.org/10.7554/eLife.42722.016

Removing $CaT$ in model (a) does not disrupt bursting activity immediately. Notice that the amplitude of the bursts remains approximately constant over a range of $gCaT$ values. The dim red and yellow lines at $\approx 20mV$ show that the amplitudes of the spikes are different and have different dependences with $gCaT$. When the model transitions into a tonic spiking regime, the amplitude of the spikes is the same and there is only one amplitude value. This value stays constant over a range but the minimum membrane potential decreases and the overall amplitude therefore increases. The model returns to a bursting regime for values of $gCaT$ smaller than $30\%gCaT$. Notice that in model (a) the membrane potential during bursts goes below $-50mV$, unlike in the control condition. Notice that the waveform of the membrane potential changes abruptly as $gCaT$ is reduced and the models transition into a spiking regime. Model (f) is less resilient to this perturbation and this transition takes place at lower conductance values.

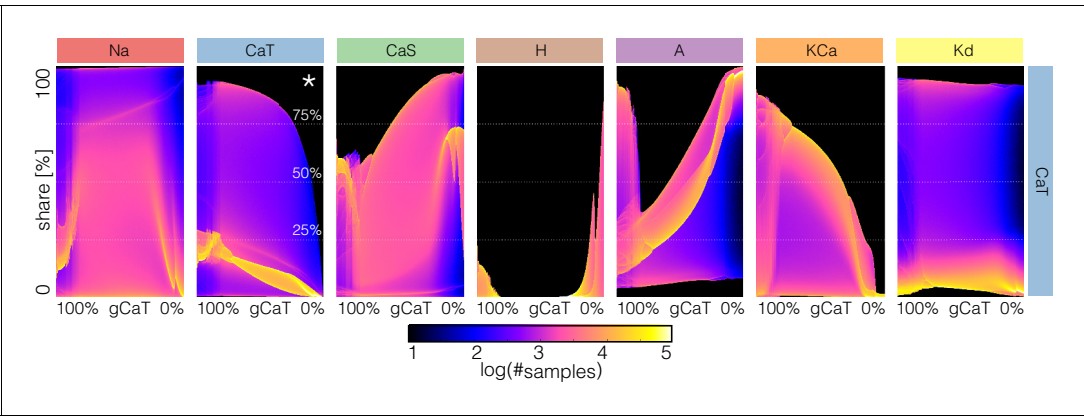

**Figure 14.** Changes in waveform of current shares as one current is gradually decreased. The panels show the probability distribution of the share of each current $\hat{C}_i(t)$ for model (f) as $CaT$ is decreased (see *Figure 14—figure supplement 1*).

DOI: https://doi.org/10.7554/eLife.42722.017

The following figure supplement is available for figure 14:

**Figure supplement 1.** Probability distributions of currents shares.

DOI: https://doi.org/10.7554/eLife.42722.018

Removing $CaS$ does not much change the waveform, but it alters the temporal properties of the activity. The models remain bursting up to a critical value and the amplitude of the spikes was changed little. The features of the slow wave do not much change either except in model (f). Model (c) is less resilient to this perturbation since it becomes quiescent for lower decrements of the maximal conductance than the other models. The effect of gradually removing $H$ appears similar to $CaS$ in this representation. In this case again, the morphology of the waveform is less altered than its temporal properties (except in model (e) where a transition takes place).

*Figure 13* shows the same plots for the outward and leak currents. The $A$ current in model (a) is very small ($gA \approx 10\mu S$) and its removal has little effect on the activity. This translates into curves that appear as parallel lines indicating spikes with different amplitudes that remain unchanged. The rest of the models exhibit a transition into a different regime. The waveforms of this regime appears similar to the waveforms which result from removing $gNa$ (see *Figure 7*) but in this representation it is easier to observe differences such as the overall amplitude of the oscillation. The amplitude decreases as $gNa$ is decreased and increases as $gA$ is decreased. Removing $KCa$ has a similar effect to removing $gCaT$ in that the models transition into tonic spiking regimes. The difference is that the spiking regimes that result from removing $KCa$ have smaller amplitudes and also correspond to more depolarized states.

All models are very sensitive to removing $Kd$ and low values result in single-spike bursting modes with large amplitudes. Model (c) is least fragile to this perturbation and exhibits a visible range ($\sim 100\%$ to $\sim 90\%$) with bursting modes. These oscillations break down in a similar way to the $Na$ case and display similar patterns. However, an important difference is that unlike in the $gNa$ case, the overall amplitude of the oscillation increases as $gKd$ is decreased. As before, the top edge corresponds to the amplitude of the large spike and the curves in the colored region correspond to extrema of the oscillation. After spiking, the membrane remains at a constant depolarized value ($\approx -20mV$) for a long period and produces a high-frequency oscillation before hyperpolarization. The amplitude of this oscillation increases as $Kd$ is further decreased, and this results in a white curve that starts above $0mV$ and ends above $0mV$. The beginning of this curve corresponds to a high-frequency oscillation that occurs after spike termination. This type of activity is termed plateau oscillations and was reported in models of leech heart interneurons (*Cymbalyuk and Calabrese, 2000*) and in experiments in lamprey spinal neurons (*Wang et al., 2014*). These features are hardly visible in the traces in *Figure 8* and are highlighted by this representation. Finally, the *Leak* case appears similar to mixture of the $Na$ and $A$ cases. The cells remain bursting over a range of values and some of them transition into a single-spike bursting mode that is different from the $KCa$ case.

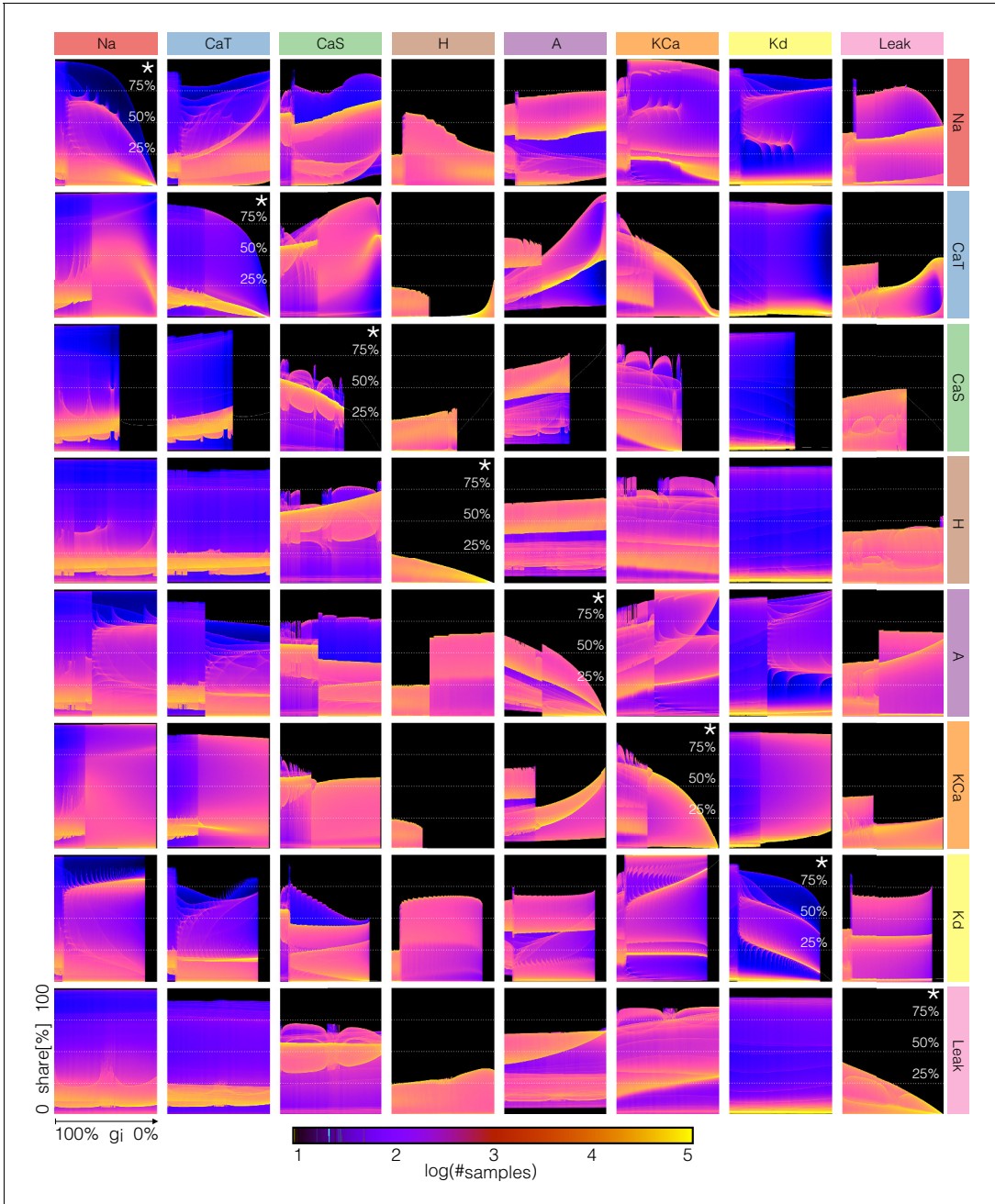

**Figure 15.** Changes in waveform of current shares as each current is gradually decreased. The panels show the probability distribution of the share of each current $\hat{C}_i(t)$ for model (c) as each current is decreased.
DOI: https://doi.org/10.7554/eLife.42722.019

## Changes in current contributions as conductances are gradually decreased

The key to the visualization method in *Figures 12* and *13* is to consider $V(t)$ not as a time series but as a stochastic variable with a probability distribution (see *Figure 3* and supplement). The same procedure can be applied to the time series of each current. However, because the contributions of the currents are different at different times, and at different decrements of conductance values, it is not possible to display this information using the same scale for all channels. To overcome this, we proceed as in the currentscapes and instead focus on the normalized currents or shares to the total

inward and outward currents (the rows of matrices $\hat{C}^+$ and $\hat{C}^-$, see Materials and methods). The current shares $\hat{C}_i(t)$ correspond to the width of the color bands in the currentscapes and can also be represented by a time series that is normalized to the interval $[0, 1]$. The probability distribution of $\hat{C}_i(t)$ permits displaying *changes* in the contributions of each current to the activity as one current is gradually removed. Interpreting these distributions is straightforward as before: the number of times the system is sampled in a given current share configuration is proportional to the time the system spends there. The aim of plotting these distributions is to visualize how the currentscapes would change for all values of the conductance decrement. To illustrate this procedure, we return to $CaT$ to explore further the causes of the complex behavior of model (f) (see *Figure 9*).

*Figure 14* shows the probability distributions of the current shares as $CaT$ is gradually decreased in model (f) (see also *Figure 9* and *Figure 14—figure supplement 1*). The panels show the share of each current as $CaT$ is gradually decreased and the probability is indicated in colors. In control the $Na$ and $CaT$ current shares are distributed in a similar way. Both currents can at times be responsible for $\approx 90\%$ of the inward current, but most of the time they contribute $\approx 20\%$. The $Na$ current is larger right before spike repolarization and the $CaT$ amounts to $\approx 90\%$ of the small ($\approx 5nA$) total inward current. For larger decrements, the system transitions into tonic spiking and the contribution of the $Na$ current is more evenly distributed over a wider range. The contribution of the $CaT$ current is predominantly $\approx 15\%$ and trends to zero as $gCaT \rightarrow 0$. Note also that as the contribution of $CaT$ decreases, the contribution of $CaS$ increases to values larger than $75\%$ while in control it contributes with $\approx 50\%$. The contribution of the $H$ current is small ($\leq 25\%$) between $100\%gCaT$ and $\approx 80\%gCaT$; it becomes negligible between $\approx 80\%gCaT$ and $\approx 20\%gCaT$ and becomes dominant after $20\%gCaT$. The $A$ current behaves similarly to the $H$. It contributes $\approx 90\%$ of the (small $\approx 2nA$) total outward current before burst initiation and its contribution decreases drastically when the system transitions into tonic spiking. As $CaT$ is removed further the $A$ current is more likely to contribute with a larger share. The contribution of the $KCa$ current decreases as $gCaT$ is decreased and some of it persists even when $gCaT$ is completely removed. In contrast, the contribution of the $Kd$ current does not appear to change much and nor does its role in the activity.

Performing the same analysis for all conductances results in a large amount of information. Despite this and because we are plotting the normalized currents or current shares, our representation allows us to display this information in a coherent fashion. As an example, in *Figure 15* we show the effect of gradually decreasing each current on all the currents in model (c). The rows indicate which conductance is decreased and the columns show the effect of this perturbation on the corresponding current. The first row shows how the shares of each current change as the $Na$ current is decreased. For instance, the effect of decreasing $gNa$ on the $Na$ current (indicated by *) is as expected, with the maxima of the distribution trending to zero as $gNa \rightarrow 0$. The effect of removing $gNa$ on the other currents is non-trivial and is displayed along the same row. Notice that while the effect of removing a current on that same current (diagonal panels) is relatively predictable, the rest of the currents become rearranged in complicated ways.

Again, a full description of these diagrams is beyond the scope of this work so we will only make some observations. When the pertubations are negligible or weak ($100\%$ to $\approx 90\%$) all currents play a role because there are periods of time in which they contribute to at least $\approx 20\%$ of the total current. There are ranges of the conductances over which small changes result in smooth transformations of the current configuration, there are specific values at which sharp transitions take place, and these values are different depending on the current that is decreased. While some of this information can also be extracted from *Figures 12* and *13*, the diagrams in *Figure 15* show how the currents get reorganized at these transitions. In addition, this arrangement is convenient for comparing the effect of decreasing each conductance on a given current. For example, the contributions of the $Na$ and $Kd$ currents change little for most perturbations (except when these conductances are decreased). In contrast, the contributions of $CaT$, $CaS$, $H$, $KCa$, and the *leak* change more noticeably. Finally, the contribution of the $A$ current increases for most conductance decrements of any type, except at the transition values where it can grow or shrink in an abrupt manner.

# Discussion

There is an ever larger availability of experimental data to inform detailed models of identified neuron types (*McDougal et al., 2017*). Experimenters have determined the kinetics of many channel types, both in vertebrate and invertebrate neurons. There are also model databases with thousands of parameters which permit the development of large scale models of neural tissue (*Bezaire et al., 2016*). One difficulty in ensemble modeling is the necessity of incorporating the biological variability observed in some of the parameters – such as the conductances – at the same time that we require the models to capture some target activity. In other words, we may be interested in modeling a type of cell that displays some steotypical behavior, and would like to obtain many different versions of such models. Two main approaches to this problem were introduced in the past. One consists of building a database of model solutions over a search domain and screening for target solutions: this considers all possible value combinations within an allowed range up to a numerical resolution and then applies quantitative criteria to determine which solutions correspond to the target activity (*Prinz et al., 2004*). An alternative approach consists of designing a target function that assigns a score to the models' solutions in such a way that lower scores correspond to solutions that meet the targets, and then optimizing these functions (*Achard and De Schutter, 2006*; *Druckmann et al., 2007*; *Ben-Shalom et al., 2012*).

Both approaches have advantages and shortcomings. In the case of the database approach, trying all posible parameter combinations in a search range becomes prohibitively expensive as more parameters are allowed to vary. One advantage of this approach is that it provides a notion of how likely it is to find conductances within a search range that will produce the activity. In the landscape approach, we find solutions by optimization and – without further analysis – we do not know how likely a given solution type is. This approach has the advantage that it can be scaled to include large numbers of parameters. Additionally, if a particular solution is interesting, we can use genetic algorithms on successful target functions to 'breed' as many closely related models as desired. Ultimately, any optimization heuristic requires blind testing random combinations of the parameters, and developing quantitative criteria for screening solutions in a database results in some sort of score function, so the two approaches are complementary. A successful target function can determine if a random perturbation results in disruption of the activity and this can be used to perform population-based sensitivity analyses (*Devenyi and Sobie, 2016*).

Regardless of the optimization approach, most work is devoted to the design of successful target functions. Different modeling problems require different target functions (*Roemschied et al., 2014*; *Fox et al., 2017*; *Migliore et al., 2018*) and one challenge in their design is that sometimes we do not know a priori if the model contains solutions that will produce good minima. In addition, a poorly constrained target function can feature multiple local minima that could make the optimization harder, so even if there are good minima they may be hard to find. One difference between the landscape functions in *Achard and De Schutter (2006)* and the ones utilized here is that in their setup model solutions are compared to a target *time series* via a phase-plane method. The functions introduced in this work use an analysis based on Poincaré sections or thresholds to characterize the waveform and to define an error or score. Instead of targeting a particular waveform, we ask that some features of the waveform – such as the frequency and the burst duration – are tightly constrained, while other features – such as the concavity of the slow waves – can be diverse. This is motivated by the fact that across individuals and species, the activity of the pyloric neurons can be diverse but the neurons always fire in the same sequence and the burst durations have a well-defined mean. Our approach is successful in finding hundreds of models that display a target activity in minutes using a commercially available desktop computer. Application of evolutionary techniques to optimize these functions provides a natural means to model the intrinsic variability observed in biological populations.

One of the main benefits of computational modeling is that once a behavior of interest is successfully captured we then possess a mechanistic description of the phenomena that can be used to test ideas and inform experiments (*Coggan et al., 2011*; *Lee et al., 2016*; *Devenyi and Sobie, 2016*; *Gong and Sobie, 2018*). As the models gain biophysical detail these advantages wane in the face of the complexity imposed by larger numbers of variables and parameters. Conductance-based models of neural activity generate large amounts of data that can be hard to visualize and interpret. The development of novel visualization procedures has the potential to assist intuition into the details of

how these models work (*Gutierrez et al., 2013*). Here, we introduced a novel representation of the dynamics of the ionic currents in a single compartment neuron. Our representation is simple and displays in a concise way the contribution of each current to the activity. This representation is easily generalizable to multi-compartment models and small networks, and to any type of electrically excitable cell, such as models of cardiac cells (*Britton et al., 2017*).

We employed these procedures to build many similar bursting models with different conductance densities and to study their response to perturbations. The responses of the models to current injections and gradual decrements of their conductances can be diverse and complex. Inspection of the ISI distributions revealed wide ranges of parameter values for which the activity appears irregular, and similar regimes can be attained by gradually removing some of the currents. Period doubling routes to chaos in neurons have been observed experimentally and in conductance-based models (*Hayashi et al., 1982*; *Hayashi and Ishizuka, 1992*; *Szücs et al., 2001*; *Canavier et al., 1990*; *Xu et al., 2017*). The sort of bifurcation diagrams displayed by these models upon current injection are qualitatively similar to those exhibited by simplified models of spiking neurons for which further theoretical insight is possible (*Touboul and Brette, 2008*). Period doubling bifurcations and low dimensional chaos arise repeatedly in neural models of different natures including rate models (*Ermentrout, 1984*; *Alonso, 2017*). The bursters studied here are close (in parameter space) to aperiodic or irregular regimes suggesting that such regimes are ubiquitous and not special cases.

We showed that in these model neurons similar membrane activities can be attained by multiple mechanisms that correspond to different current compositions. Because the dynamical mechanisms driving the activity are different in different models, perturbations can result in qualitatively different scenarios. Our visualization methods allow us to gather intuition on how different these responses can be and to explore the contribution of each current type to the neural activity. Even in the case of single compartment bursters, the response to perturbations of a population can be diverse and hard to describe. To gain intuition into the kind of behaviors the models display upon perturbation, we developed a representation based on the probability of the membrane potential $V$. This representation permits displaying changes in the waveform of $V$ as each current is blocked. This representation shows that the models respond to perturbations in different ways, but that there are also similarities among their responses. A concise representation of the effect of a perturbation is a necessary step towards developing a classification scheme for the responses.

## Materials and methods

Numerical data and data analysis and plotting code, sufficient to reproduce the figures in the paper are available on Dryad Digital Repository (https://dx.doi.org/10.5061/dryad.d0779mb).

### Model equations

The membrane potential $V$ of a cell containing $N$ channels and membrane capacitance $C$ is given by:

$$C\frac{dV}{dt} = I_e - \sum_{i=1}^{8} I_i. \tag{5}$$

Each term in the sum corresponds to a current $I_i = g_i m^{p_i} h^{q_i}(V - E_i)$ and $I_e$ is externally applied current. The maximal conductance of each channel is given by $g_i$, $m$ and $h$ are the activation and inactivation variables, the integers $p_i$ and $q_i$ are the number of gates in each channel, and $E_i$ is the reversal potential of the ion associated with the i-th current. The reversal potential of the Na, K, H and leak currents were kept fixed at $E_{Na} = 30mV$, $E_K = -80mV$, $E_H = -20mV$ and $E_{leak} = -50mV$ while the calcium reversal potential $E_{Ca}$ was computed dynamically using the Nernst equation assuming an extracellular calcium concentration of $3 \times 10^3 \mu M$. The kinetic equations describing the seven voltage-gated conductances were modeled as in *Liu et al. (1998)*,

$$\tau_{m_i}(V)\frac{dm_i}{dt} = m_{\infty_i}(V) - m_i$$
$$\tau_{h_i}(V)\frac{dh_i}{dt} = h_{\infty_i}(V) - h_i. \tag{6}$$

The functions $\tau_{m_i}(V)$, $m_{\infty_i}(V)$, $\tau_{h_i}(V)$ and $h_{\infty_i}(V)$ are based on the experimental work of

*Turrigiano et al., 1995* and are listed in refs. (*Liu et al., 1998*; *Turrigiano et al., 1995*). The activation functions of the $K_{Ca}$ current require a measure of the internal calcium concentration $[Ca^{+2}]$ (*Liu et al., 1998*). This is an important state variable of the cell and its dynamics are given by,

$$\tau_{Ca}\frac{d[Ca^{+2}]}{dt} = -Ca_F(I_{CaT} + I_{CaS}) - [Ca^{+2}] + Ca_0. \tag{7}$$

Here, $Ca_F = 0.94 \frac{\mu M}{nA}$ is a current-to-concentration factor and $Ca_0 = 0.05\,\mu M$. These values were originally taken from Liu et al. and were kept fixed. Finally, $C = 10nF$. The number of state variables or dimension of the model is 13. We explored the solutions of this model in a range of values of the maximal conductances and calcium buffering time scales. The units for voltage are $mV$, the conductances are expressed in $\mu S$ and currents in $nA$. Voltage traces were obtained by numerical integration of *Equation 5* using a Runge-Kutta order 4 (RK4) method with a time step of $dt = 0.1msec$ (*Press et al., 1988*). We used the same set of initial conditions for all simulations in this work $V = -51mV$, $m, h_i = 0$ and $[Ca^{+2}] = 5\mu M$. For some values of the parameters, the system (*Equation 5*) can display multistability (*Cymbalyuk et al., 2002*; *Shilnikov et al., 2005*).

## Optimization of target function

Optimization of the objective function *Equation 2* is useful to produce sets of parameters g that result in bursting regimes. In this work, the optimization was performed over a search space of allowed values listed here: we searched for $g_{Na} \in [0, 2 \times 10^3]$ ($[\mu S]$), $g_{CaT} \in [0, 2 \times 10^2]$, $g_{CaS} \in [0, 2 \times 10^2]$, $g_A = 2 \times [0, 10^2]$, $g_{KCa} \in [0, 2 \times 10^3]$, $g_{Kd} \in [0, 2 \times 10^2]$, $g_H \in [0, 2 \times 10^2]$, $g_L \in [0, 2 \times 10]$, $\tau_{Ca} \in [0, 10^3]$ ($[msecs]$). We minimized the objective function using a standard genetic algorithm *Holland (1992)*. This is optimization technique is useful to produce large pools of different solutions and is routinely utilized to estimate parameters in biophysical models (see for example *Assaneo and Trevisan, 2010*). The algorithm was started with a population of 1000 random seeds that were evolved for $\approx 10000$ generations. The mutation rate was $5\%$. Fitter individuals were chosen more often to breed new solutions (elitism parameter was 1.2 with 1 corresponding to equal breeding probability). The computation was performed on a multicore desktop computer (32 threads) and takes about $\approx 1$ hr to produce good solutions.

## Currentscapes

The currentscapes are stacked area plots of the normalized currents. Although it is easy to describe their meaning, a precise mathematical definition of the images in *Figure 2* can appear daunting in a first glance. Fortunately, the implementation of this procedure results in simple python code.

The time series of the 8 currents can be represented by a matrix $C$ with 8 rows and $n_{secs} \times \frac{1}{dt} = N$ columns. For simplicity, we give a formal definition of the currentscapes for positive currents. The definition is identical for both current signs and is applied independently for each. We construct a matrix of positive currents $C^+$ by setting all negative elements of $C$ to zero, $C_{i,j}^+ = C_{i,j} \mid C_{i,j} > 0$ and $C_{i,j}^+ = 0 \mid C_{i,j} \leq 0$. Summing $C^+$ over rows results in a normalization vector $n^+$ with $N$ elements

**Table 1.** Parameters used in this study and error value.

|  | gNa | gCaT | gCaS | gA | gKCa | gKd | gH | gL | $\tau_{Ca}$ | $E(g)$ |
|---|---|---|---|---|---|---|---|---|---|---|
| model (a) | 1076.392 | 6.4056 | 10.048 | 8.0384 | 17.584 | 124.0928 | 0.11304 | 0.17584 | 653.5 | 0.051 |
| model (b) | 1165.568 | 6.6568 | 9.5456 | 54.5104 | 16.328 | 110.7792 | 0.0628 | 0.10676 | 813.88 | 0.053 |
| model (c) | 1228.368 | 7.0336 | 11.0528 | 117.5616 | 16.328 | 111.2816 | 0.13816 | 0.10676 | 605.98 | 0.027 |
| model (d) | 1203.248 | 6.6568 | 10.5504 | 59.5344 | 16.328 | 111.4072 | 0.0 | 0.10676 | 653.5 | 0.471 |
| model (e) | 1210.784 | 8.164 | 6.28 | 113.04 | 12.56 | 118.4408 | 0.1256 | 0.0314 | 393.13 | 0.109 |
| model (f) | 1245.952 | 7.7872 | 6.7824 | 84.6544 | 12.56 | 113.9192 | 0.02512 | 0.0 | 174.34 | 0.047 |
| model (*Figure 2*) | 1228.368 | 7.0336 | 11.0528 | 117.5616 | 16.328 | 110.7792 | 0.13816 | 0.10048 | 605.98 | 0.007 |
| model (*Figure 3*) | 895.528 | 3.8936 | 16.5792 | 116.4312 | 21.352 | 115.6776 | 0.0 | 0.08792 | 828.73 | 0.058 |

DOI: https://doi.org/10.7554/eLife.42722.020

$n_j^+ = \sum_i C_{i,j}^+$. The normalized positive currents can be obtained as $\hat{C}^+ = C^+/n^+$ (element by element or entry-wise product). Matrix $\hat{C}^+$ is hard to visualize as it is. The columns of $\hat{C}^+$ correspond to the shares of each positive current and can be displayed as pie charts (see *Figure 2*). Here, instead of mapping the shares to a pie we map them to a segmented vertical 'churro'. The currentscapes are generated by constructing a new matrix $C_S$ whose number of rows is given by a resolution factor $R = 2000$, and the same number of columns $N$ as $C$. Each column $j$ of $\hat{C}^+$ produces one column $j$ of $C_S$. Introducing the auxiliary variable $p_{i,j} = \hat{C}_{i,j}^+ * R$ we can define the currentscape as,

$$C_{S_{i,j}} = k \mid \sum_m^k p_{m,j} \leq i < p_{k+1,j} + \sum_m^k p_{m,j}. \tag{8}$$

The current types are indexed by $k \in [0,7]$ and we assume $\sum_m^{k=0} p_{m,j} = 0$. The black filled curve in *Figure 2B* corresponds to the normalization vector $n^+$ plotted in logarithmic scale. We placed dotted lines at $5nA$, $50nA$ and $500nA$ for reference throughout this work. The currentscapes for the negative currents are obtained by applying definition (*Equation 8*) to a matrix of negative $C^-$ currents defined in an analogous way as $C^+$. Finally, note that matrices $\hat{C}^+$ and $\hat{C}^-$ are difficult to visualize as they are. The transformation given by definition (*Equation 8*) is useful to display their contents.

### ISI distributions

We inspected the effects of injecting currents in our models by computing the inter-spike interval ISI distributions. For this, we started the models from the same initial condition and simulated them for 580 s. We dropped the first 240 s to remove transient activity and kept the last 240 s for analysis. Spikes were detected as described before. We collected ISI values for $N = 1001$ values of injected current equally spaced between $-1nA$ and $5nA$.

### V distributions

To sample the distributions of $V$ we simulated the system with high temporal resolution ($dt = 0.001msec$) for 30 s, after dropping the first 120 s to remove transients. We then sampled the numerical solution at random time stamps and kept $2 \times 10^6$ samples $V = \{V_i\}$ for each percent value. We took 1001 values between 1 and 0.

### Parameters

Model parameters used in this study are listed in *Table 1*.

## Acknowledgements

LMA acknowledges Marcos Trevisan for early discussions on genetic algorithms and Francisco Roslan for programming training.

## Additional information

#### Competing interests

Eve Marder: Senior editor, *eLife*. The other author declares that no competing interests exist.

#### Funding

| Funder | Grant reference number | Author |
| --- | --- | --- |
| National Institutes of Health | R35 NS097343 | Eve Marder |
| Swartz Foundation | 2017 | Leandro M Alonso |
| National Institutes of Health | MH046742 | Eve Marder |
| National Institutes of Health | T32 NS07292 | Leandro M Alonso |

The funders had no role in study design, data collection and interpretation, or the decision to submit the work for publication.

## Author contributions
Leandro M Alonso, Software, Formal analysis, Investigation, Visualization, Writing—original draft, Writing—review and editing; Eve Marder, Conceptualization, Resources, Supervision, Visualization, Writing—review and editing

## Author ORCIDs
Leandro M Alonso (iD) https://orcid.org/0000-0001-8211-2855
Eve Marder (iD) https://orcid.org/0000-0001-9632-5448

## Decision letter and Author response
Decision letter https://doi.org/10.7554/eLife.42722.025
Author response https://doi.org/10.7554/eLife.42722.026

# Additional files

## Supplementary files
• Transparent reporting form
DOI: https://doi.org/10.7554/eLife.42722.021

## Data availability
All data generated or analysed during this study are included in the manuscript and supporting files. Source data files have been provided for Figures 2 through 15. Data package available in Dryad: doi:10.5061/dryad.d0779mb.

The following dataset was generated:

| Author(s) | Year | Dataset title | Dataset URL | Database and Identifier |
|---|---|---|---|---|
| Alonso LM, Marder E | 2019 | Data from: Visualization of currents in neural models with similar behavior and different conductance densities | https://dx.doi.org/10.5061/dryad.d0779mb | Dryad Digital Repository, 10.5061/dryad.d0779mb |

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
