## [Decision Letter]

Thank you for submitting your article "Visualization of currents in neural models with similar behavior and different conductance densities" for consideration by *eLife*. Your article has been reviewed by Gary Westbrook as the Senior Editor, a Reviewing Editor, and three reviewers. The following individuals involved in review of your submission have agreed to reveal their identity: Alexandre Guet-McCreight (Skinner lab) (Reviewer#1); Milad Lankarany (Reviewer #2); Oliver Britton (Reviewer #3). The reviewers have discussed the reviews with one another and the Reviewing Editor has drafted this decision to help you prepare a revised submission.

Summary:

All of the reviewers were fully appreciative of the 'tools and resources' approach and thought that the visualization tool would be helpful and useful to the wider community. At the same time, there were areas for improvement. Specifically, (i) expansion of the code to make it usable by others, (ii) more elaboration on the method itself (optimization, objective function) and perhaps less on the details of the 'results' which are clear enough from the figures and given that this is a tools and resources paper, and (iii) several small fixes throughout. We regard these points as essential in the revision, but the full comments are provided below for your guidance as you revise the manuscript.

Essential revisions:

*Reviewer #1:*

In this manuscript, the authors present several novel methodologies, including (A) an optimization technique that involves the use of thresholds (i.e. Poincaré sections) as objectives, (B) a visualization technique for better plotting ion channel currents in neuron models, (C) a visualization technique for better demonstrating changes in electrophysiology at different levels of perturbations, as well as (D) combining these two visualization techniques. Specifically, the visualization techniques shown in this manuscript demonstrate very novel, intuitive, and yet surprisingly simple ways of viewing and interpreting complex high-dimensional data. Specifically, the currentscape plots are a much-needed addition to the field and facilitate what both computational and experimental researchers can interpret from models. As well, the plots highlighting changes in voltage and ISI distributions over different perturbation levels provide another surprisingly simple but rich way of viewing large amounts of data. Overall, this manuscript merits publication but may benefit from giving more background on its "landscape" optimization method, and several sections in the results can possibly be boiled down to more succinct take-away messages.

– While the Supplementary files on Dryad are an essential component of this publication, it is not straightforward for every reader to run this code to extract the parameters used in these simulations. As such, it might be helpful for readers who are not immediately intending to try out this code, to include these values in a supplementary table. Related to this, a small glossary table of the variable names might also be helpful.

– For a reader that is somewhat naïve on the topic of optimizations, it remains unclear to me how landscape optimizations differ from other types of optimizations (i.e. why is it called a "landscape" optimization?). It might be helpful to describe a bit more how landscape optimizations work relative to other types of optimizations – it sounds as though the "landscape" part simply refers to the type of target/objective function that is being used? On a related note, very few details are given regarding the "custom genetic algorithm" (subsection “Finding parameters: landscape optimization”). Would it be possible to give more details on this, as well as parameters that were used in the optimizations (e.g. mutation, crossover, population size, etc.)?

– Often throughout the paper, results from the plots are described in more detail than is necessary and this tends to derail the focus of the story. Here I am mainly referring to the more-or-less informal descriptions of what the plots look like across different models, channel mechanisms, currents, manipulations, etc. (e.g. subsection “Perturbing the models with gradual decrements of the maximal conductances” or subsection “Changes in waveform as conductances are gradually decreased”). While these observations are very interesting in themselves, the authors often do not give enough explanation (i.e. of why those observations occur) to warrant mentioning them in the first place. In fact, many of those observations are indicative of the rich amount of information obtainable from these plotting methods, but perhaps beyond the scope this paper. Ultimately, since the authors are presenting a novel visualization technique, the plots should really be able to speak more for themselves. In my opinion, the plots do indeed speak for themselves, better in fact than the results descriptions given by the authors.

– It would be nice to apply the plotting technique in Figure 3 and Figure 5 to a set of experimental data for comparison purposes. Based on the descriptions of how these plots are made, I would imagine that this could be possible, though maybe at a lower resolution(?). If so, it might be worth mentioning, especially for experimental readers. Given that there is diversity in how the electrophysiology of the different models change with removal of channel currents and current injection amplitudes, further steps following model optimization might be to see which models can capture the changes (or diversity in changes) seen in electrophysiology. This might further help narrow the list of possible optimization solutions that can viably capture experimental electrophysiologies.

*Reviewer #2:*

This paper presents a comprehensive study on how a model neuron with different maximal conductances show similar membrane activity. I think this paper is well designed and presented. The problem of having identical observation (experimental data) given different parameter settings of a representative model is important in neuroscience. This paper introduces a visualization method to track the dynamics of ionic currents underlying each set of parameters. I have two major points to enhance the quality of the paper. Shortly, my major points are about the consistency of the results of the paper with respect to (i) objective function (total Error) and (ii) type of stimulus.

Elaborate on the effect of objective function on your results (overall) – specifically, if more detailed objective functions (or the same as yours but with different weights) might better distinguish between bursting models (which might not be inspected visually). And, I think the value of objective function (total error, Equation 4) should be reported. For the 6 selected models of bursts, you should report those values.

My second point is more like a question. What happened to the response of, for example, two models (a and b) given an identical but noisy injected current? Are they still the same?

*Reviewer #3:*

This is an interesting manuscript that describes several novel visualization techniques for interpreting the large amount of data that are routinely generated by computational models of neurons and other electrically excitable cells. The authors focus on methods to visualize how the balance of all ionic currents in a model, as well as the voltage, change when the conductance of one current is varied. This is a key part of understanding these highly non-linear models, as changes in one current often have unexpected knock on effects on the behavior of many other currents. These techniques are illustrated using six models with similar control behavior but different conductance values.

I particularly like that the techniques the authors describe allow both qualitative (e.g. firing pattern) and quantitative (e.g. peak AP voltage) changes in model output under parameter variation to be visualized within a single plot.

The authors illustrate the use of these techniques by analyzing an eight current single-compartment neuron model. I appreciate the use of a reasonably complex model as an example, rather than a toy model with few currents, as it better illustrates to readers how they might use the techniques in their own research.

My major criticism of the manuscript is that while I commend the authors for providing well organized source code that is sufficient to reproduce all figures, the study has been submitted as a tools and resources paper, and the code as is does not form a tool for other researchers to use, as it is a series of non-reusable scripts.

To allow other researchers to easily use these techniques, I would like to see the visualization code repackaged as a Python module encapsulating the main plotting functions (i.e. currentscapes; voltage probability distribution and ridge plots; ISI distribution plots; and conductance against current plots) with a documented interface. Building such a module in Python is relatively simple. I would encourage the authors to look at the Python visualization module Seaborn (https://seaborn.pydata.org/) if they are not familiar with it, to see a good example of an open source scientific visualization library.

With such a module, the code for a new user to produce, for example, a currentscape plot could be reduced to a few lines of code, e.g.:

> import current_visualization_module

> # Load pre-existing arrays of data from a simulation including currents,

> voltage, time arrays = load(path_to_data)

> current_names = ['INa', 'ICaT', 'IKA',…etc]

> # Perform the plot

> current_visualization.plot_currentscape(arrays, current_names)

Currently, most of the figure plotting code is provided as a set of scripts, but as these scripts are organized into functionally distinct sections (e.g. running the model, currentscape calculations, currentscape plotting), they could easily be converted into reusable functions. The figure plotting scripts could then be rewritten to use these functions, which would provide a gallery of examples for how to use the module. I don't think this revision would need too much extra work as most of the required code is already written and just needs to be encapsulated within a set of functions.

Releasing such a module this would substantially add to the utility and uptake of this work and provide a valuable tool for multiple electrophysiological modelling communities. From a purely selfish viewpoint, I would use it in my own work!

---

## [Author Response]

Essential revisions:Reviewer #1:In this manuscript, the authors present several novel methodologies, including (A) an optimization technique that involves the use of thresholds (i.e. Poincaré sections) as objectives, (B) a visualization technique for better plotting ion channel currents in neuron models, (C) a visualization technique for better demonstrating changes in electrophysiology at different levels of perturbations, as well as (D) combining these two visualization techniques. Specifically, the visualization techniques shown in this manuscript demonstrate very novel, intuitive, and yet surprisingly simple ways of viewing and interpreting complex high-dimensional data. Specifically, the currentscape plots are a much-needed addition to the field and facilitate what both computational and experimental researchers can interpret from models. As well, the plots highlighting changes in voltage and ISI distributions over different perturbation levels provide another surprisingly simple but rich way of viewing large amounts of data. Overall, this manuscript merits publication but may benefit from giving more background on its "landscape" optimization method, and several sections in the results can possibly be boiled down to more succinct take-away messages.– While the Supplementary files on Dryad are an essential component of this publication, it is not straightforward for every reader to run this code to extract the parameters used in these simulations. As such, it might be helpful for readers who are not immediately intending to try out this code, to include these values in a supplementary table. Related to this, a small glossary table of the variable names might also be helpful.

We fixed the supplementary data package on Dryad to make it more accessible to the general public. We also included a list of all utilized parameters in the Materials and methods section and a glossary with variable names.

– For a reader that is somewhat naïve on the topic of optimizations, it remains unclear to me how landscape optimizations differ from other types of optimizations (i.e. why is it called a "landscape" optimization?). It might be helpful to describe a bit more how landscape optimizations work relative to other types of optimizations – it sounds as though the "landscape" part simply refers to the type of target/objective function that is being used? On a related note, very few details are given regarding the "custom genetic algorithm" (subsection “Finding parameters: landscape optimization”). Would it be possible to give more details on this, as well as parameters that were used in the optimizations (e.g. mutation, crossover, population size, etc.)?

Landscape is a common way to refer to the cost function (also evolutionary landscape), so yes, the “landscape” is the function. The reason people call these functions “landscapes" is that in the case of 2 dimensions the plot of this function resembles a landscape of mountains, valleys and ridges. Optimization means finding minima of this landscape. This terminology seems fairly common (see Achard et al., 2006) and this is the reason we choose to use it.

By “custom genetic algorithm” we mean that we are using an algorithm we wrote and not an optimization package that can be referenced. We changed the word “custom" to "standard". One reason the algorithm is not described in detail is that these algorithms are very common and are sufficiently well described. But the main reason we do not discuss this much is that ultimately what matters is the function that is being optimized, and not the optimization technique itself.

We included fixes throughout the main text to better clarify these issues and discuss technical details in subsection “Optimization of target function”.

– Often throughout the paper, results from the plots are described in more detail than is necessary and this tends to derail the focus of the story. Here I am mainly referring to the more-or-less informal descriptions of what the plots look like across different models, channel mechanisms, currents, manipulations, etc. (e.g. subsection “Perturbing the models with gradual decrements of the maximal conductances” or subsection “Changes in waveform as conductances are gradually decreased”). While these observations are very interesting in themselves, the authors often do not give enough explanation (i.e. of why those observations occur) to warrant mentioning them in the first place. In fact, many of those observations are indicative of the rich amount of information obtainable from these plotting methods, but perhaps beyond the scope this paper. Ultimately, since the authors are presenting a novel visualization technique, the plots should really be able to speak more for themselves. In my opinion, the plots do indeed speak for themselves, better in fact than the results descriptions given by the authors.

We acknowledge the reviewer’s kind appreciation of our plotting techniques. Reviewer #1 points out that the two techniques in Figure 2 and Figure 3 are combined (D). We feel that this is an important point that may not be immediately clear, so we included a supplemental figure for Figure 14 (Figure 14—figure supplement 1) that shows the relationship between the currentscapes and the distributions in Figure 14. We largely modified the text in the Results section to make it more streamlined.

– It would be nice to apply the plotting technique in Figure 3 and Figure 5 to a set of experimental data for comparison purposes. Based on the descriptions of how these plots are made, I would imagine that this could be possible, though maybe at a lower resolution(?). If so, it might be worth mentioning, especially for experimental readers. Given that there is diversity in how the electrophysiology of the different models change with removal of channel currents and current injection amplitudes, further steps following model optimization might be to see which models can capture the changes (or diversity in changes) seen in electrophysiology. This might further help narrow the list of possible optimization solutions that can viably capture experimental electrophysiologies.

We agree with the reviewer in that similar plotting techniques could be applied to experimental data. However, we prefer to leave that to the reader since we did not test it and would rather not include experimental data in this work.

The reviewer points out that since the models responds differently to similar perturbations one could further narrow down the list of possible solutions by targeting models that respond to a perturbation in a prescribed way. We absolutely agree with the reviewer. There are many experimental perturbations – such as changes in temperature or pH – that result in reproducible changes in the activity. We can build similar target functions whose minima correspond to models that resemble the control data, and at the same time respond in a prescribed way to a given perturbation. We are now doing exactly this in neurons and small networks.

Reviewer #2:This paper presents a comprehensive study on how a model neuron with different maximal conductances show similar membrane activity. I think this paper is well designed and presented. The problem of having identical observation (experimental data) given different parameter settings of a representative model is important in neuroscience. This paper introduces a visualization method to track the dynamics of ionic currents underlying each set of parameters. I have two major points to enhance the quality of the paper. Shortly, my major points are about the consistency of the results of the paper with respect to (i) objective function (total Error) and (ii) type of stimulus.Elaborate on the effect of objective function on your results (overall) – specifically, if more detailed objective functions (or the same as yours but with different weights) might better distinguish between bursting models (which might not be inspected visually). And, I think the value of objective function (total error, Eq 4) should be reported. For the 6 selected models of bursts, you shall report those values.

We expanded the discussion of the cost function. More detailed cost functions (i.e. with additional targets) will result in models that satisfy more requirements. One could of course add additional restrictions like, for example, ask that the number of spikes in a burst is within a range. In this manuscript we show that, using thresholds as objectives, it is easy to design cost functions that target features of physiological relevance such as frequencies and durations of bursts. The values of the cost function for the six models studied (and the models used in Figure 2 and Figure 3) were included in a table in the Materials and methods section.

My second point is more like a question. What happened to the response of, for example, two models (a and b) given an identical but noisy injected current? Are they still the same?

We did not explore the effect of injecting noise in these models. These models are extremely complex from a mathematical standpoint and their response to stochastic stimuli is well beyond the scope of this work. However, on a first principle basis we believe the response to the referee’s question is no. If the same noisy input is injected on models (a) and (b) the responses will most likely be different. This is because the models respond differently to identical perturbations, and there is no reason to assume it won’t be the case if the perturbation is stochastic.

Reviewer #3:This is an interesting manuscript that describes several novel visualization techniques for interpreting the large amount of data that are routinely generated by computational models of neurons and other electrically excitable cells. The authors focus on methods to visualize how the balance of all ionic currents in a model, as well as the voltage, change when the conductance of one current is varied. This is a key part of understanding these highly non-linear models, as changes in one current often have unexpected knock on effects on the behavior of many other currents. These techniques are illustrated using six models with similar control behavior but different conductance values.I particularly like that the techniques the authors describe allow both qualitative (e.g. firing pattern) and quantitative (e.g. peak AP voltage) changes in model output under parameter variation to be visualized within a single plot.The authors illustrate the use of these techniques by analyzing an eight current single-compartment neuron model. I appreciate the use of a reasonably complex model as an example, rather than a toy model with few currents, as it better illustrates to readers how they might use the techniques in their own research.My major criticism of the manuscript is that while I commend the authors for providing well organized source code that is sufficient to reproduce all figures, the study has been submitted as a tools and resources paper, and the code as is does not form a tool for other researchers to use as it is a series of non-reusable scripts.To allow other researchers to easily use these techniques, I would like to see the visualization code repackaged as a Python module encapsulating the main plotting functions (i.e. currentscapes; voltage probability distribution and ridge plots; ISI distribution plots; and conductance against current plots) with a documented interface. Building such a module in Python is relatively simple. I would encourage the authors to look at the Python visualization module Seaborn (https://seaborn.pydata.org/) if they are not familiar with it, to see a good example of an open source scientific visualization library.With such a module, the code for a new user to produce, for example, a currentscape plot could be reduced to a few lines of code, e.g.:> import current_visualization_module> # Load pre-existing arrays of data from a simulation including currents, voltage, > time arrays = load(path_to_data)> current_names = ['INa', 'ICaT', 'IKA',…etc]> # Perform the plot> current_visualization.plot_currentscape(arrays, current_names)Currently, most of the figure plotting code is provided as a set of scripts, but as these scripts are organized into functionally distinct sections (e.g. running the model, currentscape calculations, currentscape plotting), they could easily be converted into reusable functions. The figure plotting scripts could then be rewritten to use these functions, which would provide a gallery of examples for how to use the module. I don't think this revision would need too much extra work as most of the required code is already written and just needs to be encapsulated within a set of functions.Releasing such a module this would substantially add to the utility and uptake of this work and provide a valuable tool for multiple electrophysiological modelling communities. From a purely selfish viewpoint, I would use it in my own work!

We followed the suggestions of the reviewer and we now provide a python module encapsulating the main plotting functions. Using this module, Figure 2 can be reproduced with a few lines,

> import currents_visualization

> pathtovoltagetrace='./numerical-data-txt/model.Figure 2.voltage.txt'

> pathtocurrents='./numerical-data-txt/model.Figure 2.currents.txt'

> voltagetrace = genfromtxt(pathtovoltagetrace)

> currents=genfromtxt(pathtocurrents)

> currents_visualization.plotCurrentscape(voltagetrace, currents)

We agree with the reviewer that this module may be more reusable than the code we used to generate the figures in the manuscript. However, we also believe that different models may require different color schemes, some modelers prefer a different ordering for the currents, and so forth. We do not intend to provide a closed-form solution such as seaboard, that is intended for much more general purposes than the types of plots discussed here.

We anticipate that these plots will have to be adapted to the details of the problem being studied, such as model definition, perturbation type, etc. We intend to provide code that is easy to read, understand, and modify, rather than it being only efficient. For this reason, the code/data package in this resubmission includes both the exact scripts that reproduce the figures and also the encapsulated python module suggested by the reviewer together with usage examples.